# Memento No More: Coaching AI Agents to Master Multiple Tasks via Hints Internalization

**Minttu Alakuijala[1],*    Ya Gao[1],*    Georgy Ananov[2],†    Samuel Kaski[1,3]**
**Pekka Marttinen[1]    Alexander Ilin[4]    Harri Valpola[4]**

[1]*Department of Computer Science, Aalto University*

[2]*Capalo AI*

[3]*Department of Computer Science, University of Manchester*

[4]*System 2 AI*

*Equal contribution.
†Work done while at Aalto University.

minttu.alakuijala@aalto.fi, ya.gao@aalto.fi

**Reviewed on OpenReview:** https://openreview.net/forum?id=AtCURrC3XA

## Abstract

As the general capabilities of artificial intelligence (AI) agents continue to evolve, their ability to learn to master multiple complex tasks through experience remains a key challenge. Current LLM agents, particularly those based on proprietary language models, typically rely on prompts to incorporate knowledge about the target tasks. This approach does not allow the agent to *internalize* this information and instead relies on ever-expanding prompts to sustain its functionality in diverse scenarios. This resembles a system of notes used by a person affected by anterograde amnesia, the inability to form new memories. In this paper, we propose a novel method to train AI agents to incorporate knowledge and skills for multiple tasks without the need for either cumbersome note systems or prior high-quality demonstration data. Our approach employs an iterative process where the agent collects new experiences, receives corrective feedback from humans in the form of hints, and integrates this feedback into its weights via a context distillation training procedure. We demonstrate the efficacy of our approach by implementing it in a Llama-3-based agent that, after only a few rounds of feedback, outperforms advanced models GPT-4o and DeepSeek-V3 in tasksets requiring correct sequencing of information retrieval, tool use, and question answering.[1]

## 1 Introduction

Rapid advancement of artificial intelligence has led to the development of AI agents powered by large language models (LLMs) (Brown et al., 2020; Ouyang et al., 2022). Thanks to their remarkable reasoning and coding abilities, these agents are capable of performing real-world tasks by interacting with their environment (Yao et al., 2023; Zhou et al., 2023a), either through API calls or code execution. As agents operate in the environment and make mistakes, they are often tuned by human experts to improve performance. Humans observe their behavior, analyze typical mistakes, and provide additional hints and guidelines in the agent's prompt. Since it is difficult to predict which hints could be relevant at a particular step of task execution, it is

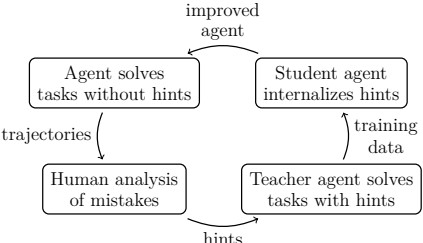

Figure 1: The proposed iterative process in which a human expert coaches an AI agent to master multiple tasks.

---

[1]Code is available at: https://github.com/minttusofia/memento-no-more.

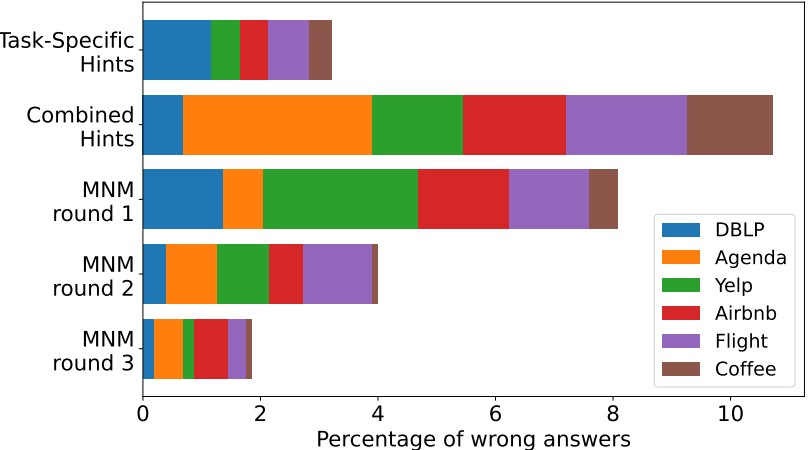

Figure 2: The percentage of wrong answers across all ToolQA test tasks. Our agent (***MNM*** for **Memento No More**) outperforms an agent that combines hints for all tasks in the prompt (*Combined Hints*), and after 3 feedback rounds even outperforms dedicated agents with only hints for the single correct task in the prompt.

```
task description
hints
inner monologue 1
python code 1
observation 1
inner monologue 2
python code 2
observation 2
...
```

Listing 1: The structure of a teacher agent prompt. The term `hints` refers to user-provided sections within the full prompt.

common to include all possible hints in the prompt. Hence, mastering multiple complex tasks requires a long and convoluted prompt which the agent must analyze exhaustively at every action step. This scenario resembles a person with anterograde amnesia who relies on a system of handwritten notes to function sensibly. As the prompt grows, the agent's performance degrades due to the overwhelming amount of information it needs to process at each step (Liu et al., 2024b). Extensive prompts are especially undesirable in Transformer-based (Vaswani et al., 2017) LLMs, as their computational cost typically scales quadratically with prompt length.

In real-world applications, AI agents need to master diverse tasks and improve by learning from their mistakes. However, relying on prompting alone, as is common in current approaches, does not seem to be a scalable solution. In this paper, we propose an algorithm for coaching an AI agent to continuously improve by incorporating feedback from a human expert. This guidance comes in the form of hints that direct the agent to perform better on a given set of tasks. Unlike in current systems, where hints are added to the prompt, we internalize these hints into the weights of the LLM through a training procedure. The proposed coaching process is iterative: after every round of training, a human expert observes the behavior of the trained agent, provides additional hints that are relevant at specific steps of the task execution, and these hints are then further internalized into the weights, and the coaching rounds continue. This iterative approach, shown in Fig. 1, allows the agent to progressively refine its understanding and execution of tasks, reducing reliance on extensive prompts. Fig. 2 illustrates our key finding: the agent trained with the proposed procedure over three rounds progressively improves its performance, despite receiving no task-specific guidance in its prompt.

## 2 Agent architecture

Our agent has a common ReAct architecture (Yao et al., 2023). Given a task description and an initial set of hints, the agent generates actions consisting of two parts: a reasoning trace (inner monologue) and an executable code snippet, together building up a trajectory as shown in Listing 1. At each step, the current trajectory forms the agent's prompt. The prompt's sections are delineated with XML tags, such as `<inner_monologue>...</inner_monologue>` for the reasoning traces. To increase compliance with the correct response format, the expected structure is outlined in a separate message immediately before requesting an inner monologue or code response. In addition, the response is initiated with the relevant opening XML tag. Similar to previous works (e.g., Wang et al. 2024a), we define the executable part of the agents' actions as

(a) **Round 1**: Internalizing generic guidance and tools description to learn novel tasks

(b) **Round 2** and onwards: Internalizing targeted hints to mitigate mistakes

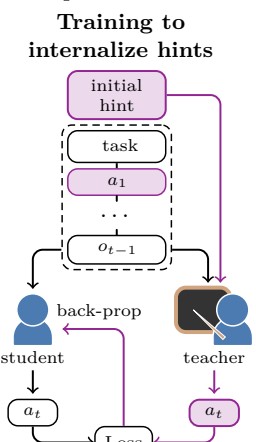

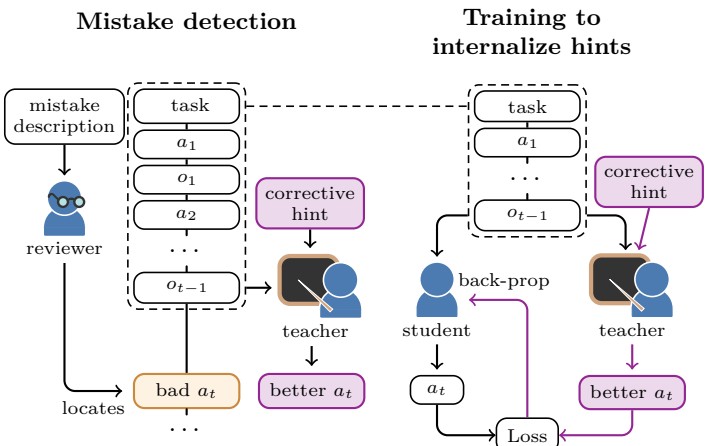

Figure 3: Overview of the proposed training procedure. **Left:** In Round 1, a student agent is trained to internalize initial hints (tool documentation and general best practices) by learning to distill the outputs (actions $a_t$) of a teacher agent that has access to these hints, while the student itself only sees a minimal task description and the execution history (actions $a_{1:t-1}$, environment observations $o_{1:t-1}$). **Right:** In Round 2 and subsequent rounds, the behavior of the trained student is further refined based on human feedback in the form of mistake descriptions and corrective hints, each addressing a specific kind of mistake the agent exhibits. These hints can be inserted only at steps where this mistake occurs, by implementing an automated reviewer (AI or a script) that locates the mistake in past trajectories. The student is then further trained to distill the outputs of a teacher that is conditioned on this corrective feedback, while the student is not.

Python code. The agent receives the execution outputs, including the content of the standard output and standard error (stderr), as its next observation.

The agent has access to a set of tools needed to complete tasks, which it invokes as Python functions within its code actions. During the first training round, these tools are documented in the initial hints. While individual tasks may require a specific subset of tools, every task execution must be finalized by calling the function `complete_task(report, answer)`, through which the agent submits the final report and the requested answer. After the first round of training, the agent internalizes the knowledge of the tools and the tool descriptions are removed from the hints.

# 3 Agent training

We now present an iterative algorithm to coach AI agents to master their tasks.

## 3.1 Round 1: Internalizing initial hints on tools and other instructions for novel tasks

**Prompt engineering for initial hints:** Initially, an AI agent is tasked with solving a set of training tasks with minimal guidelines – only the tool descriptions and formatting instructions. During this phase, we observe the agent's behavior, identify common mistakes, and introduce additional guidance and hints to improve decision-making in difficult scenarios. The hints may take various forms, such as in-context examples of tool usage, step-by-step task-solving strategies, and inner monologue recommendations. We usually provide only the minimal amount of tips necessary for successfully completing a given task and reuse the same hints for tasks of the same type. This phase effectively serves as prompt engineering, where we iteratively refine the agent's prompt to elicit the desired behavior. As with any prompt engineering steps, these improvements may generalize across different LLMs, but might require additional tuning per model. We denote the initial hints as $h_1$.

**Internalizing initial hints:** Once the initial hints have been optimized, we sample training trajectories generated by the agent with access to $h_1$, and collect these into a dataset of state-action-hint triplets, $\mathcal{D}_1 = \{(s, a, h_1(s)\}$ (see Fig. 3a). Each state $s$ in $\mathcal{D}_1$ consists of a sequence of observations and actions: $s_t = (o_0, a_1, o_1, \ldots, o_{t-1})$ within the same trajectory, where $o_0$ denotes the task description. The action $a_t$ taken in the state $s_t$ includes an inner monologue and an executable Python code. $h_1(s)$ represents the task-specific hints used in a given trajectory. The LLM that we use as the agent is a standard instruction-tuned LLM and we denote its parameters by $\theta_1$.

Next, we update the parameters of the LLM: $\theta_2 \leftarrow \theta_1 + \Delta\theta_1$, such that the new model $\theta_2$ acts as if it had access to hint set $h_1(s)$, but without needing it explicitly in each prompt:

$$\pi_{\theta_2}(a \mid s) \approx \pi_{\theta_1}(a \mid s, h_1(s)), \quad \forall (a, s, h_1(s)) \in \mathcal{D}_1.$$

Thus, the model internalizes the hints $h_1(s)$ in its weights using context distillation (Snell et al., 2022; Kujanpää et al., 2024). $\theta_2$ is constructed by adding a LoRA adapter $\Delta\theta_1$ (Hu et al., 2021) to the current weights $\theta_1$. The training set $\mathcal{D}_1$ is used to train the new model $\theta_2$, which we call the *student*, to mimic the outputs of the current model $\theta_1$, the *teacher*, without seeing the hints $h_1(s)$ in its prompt. Specifically, $\theta_2$ is trained by minimizing the KL divergence between the student's output distribution for each token in the action sequences (including reasoning and Python code) and that of the teacher's, for actions in the training set trajectories.

While we train the agent to improve its policy on a set of specific tasks, we take measures to prevent degradation of its general skills such as paying attention to its prompt. Therefore, instead of dropping out the initial hints $h_1(s)$ completely from the student's prompt, we do so with probability $p$, applied separately to each section of the hints $h_1(s)$, where $p$ is a hyperparameter, empirically set to 0.9.

### 3.2 Round $2, \ldots, I$: Internalizing corrective hints

Although Round 1 typically yields an agent that can solve tasks without any task-specific hints in its prompt, the agent often exhibits suboptimal behavior and makes mistakes in some situations. Such failure cases can be difficult to anticipate and exhaustively cover in the initial hints, so we propose a mechanism to correct for remaining issues in subsequent rounds of training: we observe the execution traces of the trained agent, identify what kind of mistakes caused failures, and provide corrective hints to improve the agent's ability to tackle similar difficult situations in the future.

We start this round by sampling trajectories with the current agent $\theta_i$ (where $i = 1, \ldots, I - 1$ is the index of the previous round) on the same set of tasks as considered in previous rounds, but without any hints. We then analyze the trajectories and design filters to automatically detect specific problems. In Fig. 3b, filtering is represented by the Reviewer, which inspects the collected trajectories. Each filter can be implemented as a script that detects certain actions or outputs (e.g., through string matching or checks on a generated program's abstract syntax tree) or they can involve calling an LLM to detect a problematic action given the description of a mistake. We design multiple filters to detect specific mistake types, and collect a set of states $\mathcal{S}_i = \{s\}$ in which the current agent makes mistakes.

Next, we seek to improve the agent's behavior by providing corrective hints $h_i(s)$ for states $s \in \mathcal{S}_i$, such that the $h_i(s)$ correct the agent's action in the problematic states. An example of a mistake, a hint and the corrected execution is shown in Fig. 4. Corrective hints are more targeted than initial hints: $h_i(s)$ depends on state $s$ to emphasize that the hints are generally state-specific. However, the same hint is applied to all states with the same mistake, such that the number of hints stays small compared to the number of states with mistakes.

We then insert the hints $h_i(s)$ at the end of the LLM prompt, after the corresponding state $s$, and sample corrected actions conditioned on the corrective hints: $a \sim \pi_{\theta_i}(a \mid s, h_i(s))$. The actions are added to a new dataset of state-action-hint triplets $\mathcal{D}_i = \{(s, a, h_i(s)\}$ used in the next round of training. Note that, as the number of detected mistakes can be small, it usually makes sense to sample multiple actions from the same state $s$ to increase the size and diversity of $\mathcal{D}_i$. It is also beneficial to balance the dataset $\mathcal{D}_i$ such that different types of mistakes and tasks are well represented (details on the data balancing strategy are

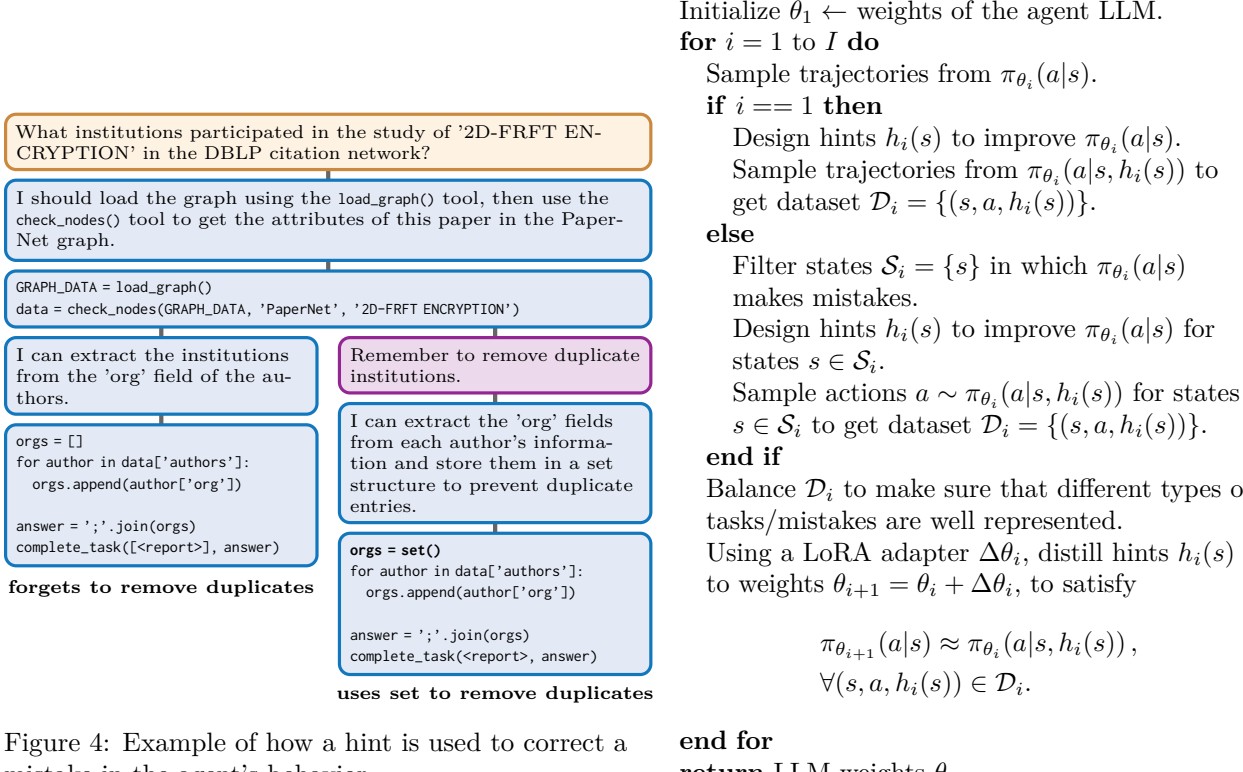

Figure 4: Example of how a hint is used to correct a mistake in the agent's behavior.

**Algorithm 1** Proposed training algorithm

Initialize $\theta_1 \leftarrow$ weights of the agent LLM.
**for** $i = 1$ to $I$ **do**
  Sample trajectories from $\pi_{\theta_i}(a|s)$.
  **if** $i == 1$ **then**
    Design hints $h_i(s)$ to improve $\pi_{\theta_i}(a|s)$.
    Sample trajectories from $\pi_{\theta_i}(a|s, h_i(s))$ to get dataset $\mathcal{D}_i = \{(s, a, h_i(s))\}$.
  **else**
    Filter states $\mathcal{S}_i = \{s\}$ in which $\pi_{\theta_i}(a|s)$ makes mistakes.
    Design hints $h_i(s)$ to improve $\pi_{\theta_i}(a|s)$ for states $s \in \mathcal{S}_i$.
    Sample actions $a \sim \pi_{\theta_i}(a|s, h_i(s))$ for states $s \in \mathcal{S}_i$ to get dataset $\mathcal{D}_i = \{(s, a, h_i(s))\}$.
  **end if**
  Balance $\mathcal{D}_i$ to make sure that different types of tasks/mistakes are well represented.
  Using a LoRA adapter $\Delta\theta_i$, distill hints $h_i(s)$ to weights $\theta_{i+1} = \theta_i + \Delta\theta_i$, to satisfy

$$\pi_{\theta_{i+1}}(a|s) \approx \pi_{\theta_i}(a|s, h_i(s)),$$
$$\forall (s, a, h_i(s)) \in \mathcal{D}_i.$$

**end for**
**return** LLM weights $\theta_I$.

provided in Appendix G). After these steps, we perform the next round of training to internalize the corrective hints $h_i(s)$. As in Round 1, training is implemented by adding a new LoRA adapter to the LLM weights $\theta_{i+1} = \theta_i + \Delta\theta_i$ and tuning the student model $\theta_{i+1}$ to mimic responses of the teacher empowered by hints $h_i(s)$. In practical scenarios, this round can be repeated multiple times to refine the agent's policy and correct any remaining mistakes. We formally present the proposed training procedure in Algorithm 1.

### 3.3 Connection to imitation learning

The proposed training algorithm is closely related to imitation learning, particularly the DAgger (Dataset Aggregation) algorithm (Ross et al., 2011). DAgger addresses the compounding error problem in behavioral cloning (Pomerleau, 1988), which arises when the learned policy encounters states not present in the expert demonstrations. When executing the policy, the learner inevitably makes mistakes, leading to deviations from the expert trajectories and entering unfamiliar states where it lacks supervision. To mitigate this issue, DAgger iteratively improves the policy by alternating between executing the current policy and querying an expert for corrective actions in the states the policy visits. These newly labeled states are added to the original expert dataset, and the policy is retrained on the aggregate dataset. By progressively incorporating expert feedback on states induced by the learner's mistakes, DAgger achieves better generalization and robustness to distribution shift.

In the proposed algorithm, the role of the expert is played by an LLM empowered by human-written hints (the teacher in Fig. 3). This eliminates the need for explicit expert demonstrations, which can be challenging to obtain. Instead, hints provide a lightweight yet effective way to generate reasonable target signals across the various states encountered during task execution. Furthermore, hints offer an additional advantage: they provide access to the full policy of the expert, represented as an output distribution over tokens, rather than just a single demonstration. This richer training signal contains significantly more information than conventional expert demonstrations. Similar to DAgger, our algorithm iteratively trains the learner's policy, observes mistakes made by the learner after each training round, and addresses them with corrective hints

that serve as expert supervision. This iterative process enables continuous improvement, leveraging the strengths of human-designed hints and the flexibility of LLMs to work in dynamic and complex environments.

## 4 Related work

**Reasoning with language models**   Prior works on eliciting reasoning in LLMs have proposed to decompose reasoning tasks into unit steps (Nye et al., 2021; Wei et al., 2022; Zhou et al., 2023b; Yao et al., 2024; Zhang et al., 2024b; Hernández-Gutiérrez et al., 2024) so that the deployed computation can scale with the complexity of the task. Generating code as a structured format to support reasoning is another common approach (Nye et al., 2021; Gao et al., 2023; Wang et al., 2024a). To improve reasoning through experience, a line of work has produced self-critiques by prompting the model to reflect on its past behavior (Shinn et al., 2024). These reflections resemble our corrective hints, but are bounded by the agent's ability to spot its own mistakes and self-correct, a known limitation (Huang et al., 2023); we instead leverage efficient human feedback for more accurate corrections.

**Human feedback in LLM training**   Multiple frameworks have incorporated human feedback in LLM training, e.g., by ranking several model answers (Ouyang et al., 2022), or providing a list of ground rules a model should not break (Bai et al., 2022).

**Interactive LLM frameworks**   Several prior works embed LLMs in interactive environments (Yao et al., 2023; Zhou et al., 2023a; Shinn et al., 2024). The most common approach has been to leverage in-context learning and add both the list of available tools and the agent's history to the current prompt, without changing the model itself. However, LLMs can struggle with information overload when prompts are the only source of new knowledge, as we show experimentally in Section 5.2. For proprietary models that cannot be improved through training, some works aim to remedy these limitations by instead optimizing the agent framework, such as the available actions (Zhang et al., 2024a; Nguyen et al., 2024) or instructions (Wu et al., 2024), or employ Retrieval Augmented Generation (Lewis et al., 2020) systems (Majumder et al., 2023; Sarch et al., 2024; Zhao et al., 2024).

**Supervised fine-tuning for interactive settings**   Supervised fine-tuning is also used to teach pre-trained LLMs new skills, but relies on access to high-quality, usually human-generated data. Efforts to collect high-quality trajectories include Song et al. (2024). This is hard to scale, and ultimately caps performance at the skill level of the annotators. For tasks with known answers, synthetic trajectories from the agent itself can be filtered to produce training data (Yao et al., 2023; Xu et al., 2023), and we compare to this in our ablations. Sarch et al. (2024) collect fine-tuning data through corrective human feedback as in our work, but for VLM agents and on the level of abstractions (rather than trajectories), introducing more possible failure points and more elements to review than in our framework.

**Knowledge distillation**   In knowledge distillation, the outputs of a larger model or ensemble of models (the teacher) are used as targets for a smaller model (the student) to transfer knowledge or capabilities (Hinton et al., 2015). This approach is only viable if a more capable model already exists. Moreover, it is not always clear if a smaller model has the representational capacity to accurately imitate the larger model, without resorting to memorizing spurious correlations in the training data (Kim et al., 2021). In **context distillation**, in contrast, the same model is used as teacher and student, but with different inputs: the teacher model has access to additional context not shown to the student (Snell et al., 2022; Padmanabhan et al., 2023; Kujanpää et al., 2024; Qi et al., 2025). Our work is the first to extend this approach to training LLM agents for interactive tasks.

# 5 Experiments with multi-task agents

## 5.1 Evaluation benchmarks

We train our MNM (Memento No More) agents to master tasks from ToolQA (Zhuang et al., 2023) and from OfficeBench (Wang et al., 2024b). Both benchmarks evaluate an agent's ability to retrieve information, use tools, and answer questions; example tasks from each are shown in Appendix F.

**ToolQA** requires the agent to fetch relevant data from external databases of various formats, including text documents, graphs, and tabular databases. We consider six ToolQA task groups: DBLP, Agenda, Yelp, Airbnb, Flight, and Coffee. Nine tools are available to the agent, whose documentation is shown in Appendix H.1, however, solving tasks from any single group requires fewer tools (two for Agenda, five for DBLP and four for the rest). We select 1,076 tasks with 100 distinct question templates (details in Appendix I.1), divided with a 60 / 10 / 30% split into training, validation, and test tasks.

**OfficeBench** evaluates an agent's ability to perform complex office workflows across 6 applications: Email, Calendar, Word, Excel, OCR and PDF reader. The agent has access to 20 tools, whose documentation is shown in Appendix H.2. After filtering out infeasible tasks (details in Appendix I.2), 286 tasks remain, split 67 / 3 / 30% into training, validation and test. Unlike in ToolQA, OfficeBench tasks are not templated but each task is unique, making generalization more challenging. Moreover, each task requires 1–3 apps, so tool sharing and app overlap varies more flexibly per task.

For each benchmark, the goal is an agent able to handle any of the benchmark's test tasks, without information on the task's group or which tools it requires. We use Llama-3.1-70B-Instruct (Grattafiori et al., 2024) as the LLM engine of our agent (referred to as Llama-3.1-70B in the following sections). In all results tables, we report the mean success rate on test tasks across 3 trials, with standard error.

## 5.2 Baseline agents

**Single-task agent baselines:** We begin by evaluating Llama-3.1-70B performance on each benchmark in a setting where the agent receives only hints relevant to the current task. An example prompt with such *task-specific hints* is shown in Appendix J. The results in Tables 1 and 3 indicate that providing only the documentation of task-specific tools is insufficient, yielding relatively low success rates of 66.1% for ToolQA and 17.2% for OfficeBench. However, when additional task-specific explanations and tool usage examples (together referred to as *best practices*) are included, the agent's performance improves significantly, achieving success rates of 96.9% and 89.1%, respectively. This gives an estimate of the upper bound of the base Llama's performance on the benchmarks.

**Multi-task agent baselines:** We now consider the primary scenario of interest, where the agent must handle tasks from any group, in the absence of task-specific information. To implement this, we combine the task-specific hints from all task types into a single prompt, per benchmark, and tune these combined hints on tasks from the training sets. This setting results in significant success rate drops, 96.9→88.4% on ToolQA (see Table 2) and 89.1→81.3% on OfficeBench (Table 4). We refer to this phenomenon as the *Memento effect*. Notably, even larger models like DeepSeek-V3 (Liu et al., 2024a) and GPT-4o (OpenAI 2023; 2024; version `gpt-4o-2024-08-06`) struggle with the long, complicated prompt. While GPT-4o performance is the strongest of these models at 92.8% on ToolQA and 89.9% on OfficeBench, it is still below Llama's ToolQA performance when hints are filtered per task. Importantly, the combined hints were tuned on training tasks separately per model, as they may not generalize across models: model failures were inspected and new hints were added until performance plateaus. For cost reasons, we use one trial for GPT-4o, and 3 for all other models.

## 5.3 Training a multi-task agent

**Round 1:** Next, we conduct the first round of training for the Llama-based ToolQA and OfficeBench agents. To collect training data, we run the agent using the *task-specific hints* described in Section 5.2. More details on our training procedure, including hyperparameter settings, are given in Appendix D. Following Round 1, the success rate of the agent, which no longer requires any guidance in the prompt, reaches 91.8% on

Table 1: Success rates of the **Single-task agent** (Llama-3.1-70B) with task-specific hints in ToolQA.

| Hints in prompt | DBLP | Agenda | Yelp | Airbnb | Flight | Coffee | Avg. |
|---|---|---|---|---|---|---|---|
| task-specific tools | $90.4_{2.0}$ | $56.3_{3.5}$ | $64.4_{1.0}$ | $50.3_{1.3}$ | $49.4_{0.6}$ | $85.4_{2.0}$ | $66.1_{0.8}$ |
| + best practices | $93.2_{1.0}$ | $100.0_{0.0}$ | $97.2_{0.6}$ | $96.7_{0.7}$ | $96.1_{1.5}$ | $98.1_{1.2}$ | $96.9_{0.4}$ |

Table 2: Success rates of **Multi-task agents** in ToolQA.

| Agent | Hints in prompt | DBLP | Agenda | Yelp | Airbnb | Flight | Coffee | Avg. |
|---|---|---|---|---|---|---|---|---|
| Llama-3.1-70B | combined tools | $61.0_{2.9}$ | $59.5_{0.8}$ | $61.0_{0.9}$ | $51.6_{2.5}$ | $49.4_{2.2}$ | $83.1_{0.8}$ | $61.0_{0.8}$ |
| Llama-3.1-70B | combined tools + best practices | $96.0_{0.6}$ | $73.8_{2.9}$ | $91.0_{1.1}$ | $88.2_{2.0}$ | $88.3_{2.0}$ | $93.0_{0.9}$ | $88.4_{0.7}$ |
| DeepSeek-V3 | | $89.8_{2.0}$ | $73.0_{2.9}$ | $87.6_{2.5}$ | $83.0_{2.3}$ | $92.8_{2.2}$ | $98.6_{0.8}$ | $87.5_{0.9}$ |
| GPT-4o | | $98.3$ | $76.2$ | $96.6$ | $92.2$ | $93.3$ | **100.0** | $92.8$ |
| MNM round 1 | | $92.1_{2.0}$ | $94.4_{2.1}$ | $84.7_{1.2}$ | $89.5_{2.0}$ | $92.2_{1.5}$ | $97.7_{0.9}$ | $91.8_{0.7}$ |
| MNM round 2 | – | $97.7_{1.1}$ | $92.9_{1.4}$ | $94.9_{1.0}$ | $96.1_{1.1}$ | $93.3_{1.0}$ | $99.5_{0.5}$ | $95.7_{0.4}$ |
| MNM round 3 | | $\mathbf{98.9}_{0.6}$ | $\mathbf{96.0}_{0.8}$ | $\mathbf{98.9}_{1.1}$ | $\mathbf{96.1}_{0.0}$ | $\mathbf{98.3}_{0.1}$ | $99.5_{0.5}$ | $\mathbf{97.9}_{0.3}$ |

Table 3: Success rates of the Llama-3.1-70B **single-task agent** with task-specific hints in OfficeBench.

| Hints in prompt | 1 app | 2 apps | 3 apps | Avg. |
|---|---|---|---|---|
| task-specific tools | $29.0_{1.2}$ | $17.3_{1.2}$ | $5.38_{1.1}$ | $17.2_{0.4}$ |
| + best practices | $90.3_{2.1}$ | $92.6_{2.1}$ | $84.9_{4.7}$ | $89.1_{2.0}$ |

Table 4: Success rates of **Multi-task agents** in OfficeBench.

| Agent | Hints in prompt | 1 app | 2 apps | 3 apps | Avg. |
|---|---|---|---|---|---|
| Llama-3.1-70B | combined tools | $29.0_{3.2}$ | $13.6_{1.2}$ | $1.08_{1.1}$ | $14.6_{1.7}$ |
| Llama-3.1-70B | combined tools + best practices | $78.5_{5.4}$ | $87.7_{1.2}$ | $78.5_{4.3}$ | $81.3_{2.9}$ |
| DeepSeek-V3 | | $88.2_{3.9}$ | $88.9_{0.0}$ | $83.9_{1.9}$ | $86.9_{1.0}$ |
| GPT-4o | | **93.5** | $88.9$ | $87.1$ | $89.9$ |
| MNM round 1 (ours) | | $81.7_{3.9}$ | $86.4_{1.2}$ | $80.6_{6.7}$ | $82.8_{3.2}$ |
| MNM round 2 (ours) | – | $87.1_{1.7}$ | $91.4_{2.5}$ | $83.9_{1.9}$ | $87.3_{1.0}$ |
| MNM round 3 (ours) | | $89.2_{2.1}$ | $\mathbf{92.6}_{2.1}$ | $\mathbf{89.2}_{1.1}$ | $\mathbf{90.3}_{0.7}$ |

ToolQA and 82.8% on OfficeBench. While this falls short of the teacher models' 96.9% and 89.1%, the trained agents surpass the combined prompt Llama agent on both benchmarks. On ToolQA, our agent additionally outperforms DeepSeek-V3 and closely approaches GPT-4o's success rate (Tables 1–4).

**Round 2:** After Round 1, the agents exhibit several mistake patterns, including forgetting to print values for inspection, incorrectly post-processing data extracted from the database, opting for unnecessary programmatic solutions, failing to remove duplicate values, reasoning incorrectly about retrieved documents, and providing incorrect tool arguments. To address these issues, we again collect data for a second round of training by solving each training task using the latest trained agents without any hints. We design 14 filters for ToolQA and 12 for OfficeBench, each describing a particular mistake pattern. The agent steps affected by each mistake type can then be automatically detected in past agent trajectories using an LLM-as-a-judge approach, guided by the mistake description. An example prompt for these LLM-based reviewers is provided in Appendix L. Once the erroneous steps are identified, we construct a set of mistake-dependent hints, denoted as $h_2(s)$, which are inserted at the end of the agent's prompt to guide it toward improved actions. The LLM reviewers

Table 5: Success rates on **unseen tasks** in ToolQA.

| Agent | Hints in prompt | DBLP | Agenda | Yelp | Airbnb | Flight | Coffee | Avg. |
|---|---|---|---|---|---|---|---|---|
| Llama-3.1-70B | combined tools | $51.7_{3.4}$ | $62.5_{1.6}$ | $53.9_{1.9}$ | $54.2_{3.2}$ | $40.0_{2.5}$ | $87.7_{1.5}$ | $58.3_{1.0}$ |
| Llama-3.1-70B | combined tools + best practices | $91.4_{0.9}$ | $50.0_{0.9}$ | $66.1_{1.3}$ | $84.7_{1.8}$ | $63.3_{3.0}$ | $93.1_{0.7}$ | $74.7_{0.5}$ |
| DeepSeek-V3 | | $93.1_{1.0}$ | $66.1_{2.1}$ | $84.9_{3.7}$ | $49.7_{0.5}$ | $55.0_{1.9}$ | $99.0_{1.0}$ | $74.6_{0.2}$ |
| GPT-4o | | **98.3** | 68.7 | **89.1** | 89.8 | **76.7** | **100** | **87.1** |
| MNM round 1 | | $90.8_{0.6}$ | $68.7_{0.4}$ | $69.7_{1.6}$ | $89.8_{1.0}$ | $72.2_{1.5}$ | $87.3_{1.0}$ | $79.7_{0.4}$ |
| MNM round 2 | – | $96.0_{0.6}$ | $\mathbf{78.1}_{0.5}$ | $73.3_{1.6}$ | $94.4_{0.6}$ | $75.6_{1.2}$ | $97.5_{0.6}$ | $85.8_{0.4}$ |
| MNM round 3 | | $96.6_{1.0}$ | $75.0_{0.9}$ | $80.0_{1.0}$ | $\mathbf{96.6}_{0.0}$ | $73.3_{1.7}$ | $96.6_{0.5}$ | $86.4_{0.4}$ |

localized these hints to 99 unique states in ToolQA trajectories, and 66 states in OfficeBench. Multiple teacher actions are sampled in each of these states to increase the amount of training data.

After the second training round, the agents reach success rates of 95.7% on ToolQA and 87.3% on OfficeBench, without hints (see Tables 2 and 4). Notably, the trained agents, despite operating with a minimal prompt and without any external guidance, now surpass GPT-4o on ToolQA and narrowly beat DeepSeek-V3 on OfficeBench, demonstrating the effectiveness of our training methodology.

**Round 3:** After Round 2, we again conduct an analysis of typical mistakes made by the agents, identifying several recurring issues. These include failing to print values for inspection, applying incorrect post-processing steps to data extracted from the database, misinterpreting the meaning of certain columns, and not adhering precisely to the task descriptions when referencing names. To further refine the agents, we again run them on the training tasks and design 18 filters for ToolQA and 12 for OfficeBench. This process identified 68 unique ToolQA states and 30 OfficeBench states with flawed outputs, for which we design new hints $h_3(s)$, and repeat training.

The resulting ToolQA agent achieves an average success rate of 97.9%, significantly surpassing all prompting-based agents. The OfficeBench agent's success rate of 90.3% similarly exceeds all other agents, though only narrowly for GPT-4o, which our agent outperforms on 2 and 3 app tasks but underperforms on 1 app tasks. Beyond increased success rates, the trained agents also demonstrate notable advantages in inference efficiency, achieving a **3–4x improvement in inference speed** compared to the untrained Llama agent while also drastically reducing inference costs by using **only 7–10% of the input token count** required by the other models. Based on observed runtimes of 90 seconds per ToolQA task and 140 seconds per OfficeBench task for our Round 3 agents, the computational cost of all 3 rounds of training is amortized after as few as 255–285 task executions in either domain (see Appendix C for details). The computational cost of training is reported in Appendix D.3 and the human annotation time in Appendix D.4.

### 5.4 Additional experiments

**Performance on unseen ToolQA tasks:** We further test how well the MNM agent generalizes to ToolQA question templates that were not present in the training data. We hold out a subset of templates by dividing them with a 60 / 10 / 30% split into training, validation, and test sets. The results are presented in Table 5, showing that the trained agent is able to generalize to unseen questions. After the third round of training, the agent outperforms DeepSeek-V3 and narrowly trails GPT-4o in average performance in this setting. Note that in OfficeBench, no tasks share templated structure with each other, and thus evaluation is always on unseen tasks.

**Performance on standard benchmarks:** To test whether the trained agents exhibit performance degradation in other domains, we evaluate their backbone LLMs on two standard benchmarks for coding and mathematical reasoning: HumanEval (Chen et al., 2021) and GSM8K (Cobbe et al., 2021). The results in Table 6 in Appendix B.1 show no signs of performance degradation.

**Ablation with SFT loss on hint-powered trajectories:** In this experiment, we use the tokens produced by the teacher as targets for the student and minimize the cross-entropy loss, the standard loss in supervised fine-tuning (SFT). We train the student model only on trajectories in which the teacher correctly solves the task. This is analogous to the approach taken in AgentBank (Song et al., 2024), but with trajectories collected with our corrective hint-internalization pipeline with iterative training rounds, rather than collecting supervision trajectories from a more advanced LLM, which may not be available in general.

We found this alternative training procedure with cross-entropy (SFT) loss to lead to slightly worse results, as reported in Table 7 in Appendix B.2.

**Ablations on data balancing and hint dropout rate:** We further experimentally validate the individual contributions of both the data balancing step and the choice of dropout rate $p$ applied to each hint section. Results without balancing, and for different dropout rates are shown in Appendix B.3.

### 5.5 Discussion and limitations

In this work, we develop multi-task agents within a single domain, and did not consider continuing training of a ToolQA agent on OfficeBench, or vice versa. We leave exploring this for future work. Moreover, our training protocol depends on human supervision in the form of hints, which can be hard to scale. However, several measures are taken to reduce human effort: reuse of hints in similar situations, automated reviewers, and only reviewing failed trajectories. While the amount of supervision needed is likely to grow with the number and complexity of tasks, our experiments show highly annotation-efficient training, with just 32 corrective hints required for ToolQA, and 24 for OfficeBench.

The human annotators in our experiments are AI researchers and authors of this work; however, we estimate that the LLM prompting expertise level needed for applying our method is by no means prohibitive for wide adoption, and is largely representative of that of machine learning engineers, who would most likely be developing AI agent systems in practice. We document the kinds of hints and LLM reviewer prompts that an annotator might be expected to write in Appendices K and L, respectively. Further discussion on the limitations of our work is included in Appendix A.

Our training approach emphasizes maintaining human oversight and control. Deploying agents in the real world could have unintended effects. We recommend careful consideration of deployment contexts and appropriate guardrails, such as limiting access to the internet or the file system when appropriate. Anticipated broader impacts and deployment strategies are outlined in Section 6.

## 6 Conclusions

In this paper, we introduced a novel approach for training AI agents to master multiple tasks efficiently, without the need for increasingly complex and expanding prompt engineering. Our method relies on an iterative training process, where the agent internalizes human-provided hints into its model weights, reducing dependency on external prompts over time. Our experimental results demonstrate the efficiency and scalability of this strategy: starting from Llama-3.1-70B and with only three rounds of mistake correction through human feedback, we achieve a remarkable 97.9% success rate on ToolQA, and 90.3% on OfficeBench. These numbers surpass those of significantly larger models, including GPT-4o (92.8% and 89.9%, respectively) and DeepSeek-V3 (87.5% and 86.9%), highlighting the effectiveness of our approach. These findings suggest that integrating structured human feedback directly into model training can lead to superior task performance while maintaining data and inference efficiency. Future work will explore extending this methodology to broader domains and investigate its applicability in real-world tasks.

**Broader Impact Statement**

This work advances LLMs' capabilities for solving complex multi-step tasks involving information retrieval, tool use, and question answering. Our methods aim to enhance AI systems' ability to assist humans in solving challenging or laborious tasks.

**Potential benefits**  Enhanced problem-solving abilities could improve automation of complex cognitive tasks across domains like research, education, and professional services. The methods we develop could advance our understanding of how to create more capable and controllable AI systems. The tight integration of human feedback in our training strategy may lead to more reliable and verifiable AI outputs than alternative approaches emphasizing task success only. Moreover, by reducing prompting overhead, our method can reduce the energy consumption and thus environmental impact of AI agent deployment.

**Societal considerations**  AI agents could accelerate automation of knowledge work, potentially impacting employment in certain sectors. We anticipate these kinds of tools to replace some current routine knowledge work with work on guiding how such problems can be solved. This can be more meaningful work and increase efficiency, but potentially affect the number of jobs. Furthermore, important questions about transparency and accountability arise when AI systems chain together multiple tools and information sources. In addition to the potential for misuse present in existing LLM systems, more autonomous AI agents could also enable new harmful use cases, such as cyberattacks.

Unlike prompt-based approaches, knowledge distilled into model weights is also not directly interpretable or externally auditable, making it harder to verify which hints or constraints the agent is following at any given step.

**Mitigation strategies**  Our training approach emphasizes maintaining human oversight and control throughout the operation of AI agents. This includes developing intuitive interfaces for monitoring agent performance, providing clear mechanisms for intervention and correction, and establishing transparent logging of all of the agent's actions. Furthermore, our methodology explicitly integrates human feedback loops not just for initial training, but also for ongoing refinement and validation, ensuring that practitioners can iteratively guide the AI's behavior and adapt it to evolving task requirements or unforeseen circumstances.

The development of more autonomous AI systems should involve setting appropriate bounds for AI agency. We recommend careful consideration of deployment contexts and proactive definition of appropriate guardrails, such as limiting access to sensitive resources including unrestricted internet browsing or file system modifications when such capabilities are not required for the task at hand. Clear protocols for human review of critical decisions made by the AI, such as purchases, should also be established.

Finally, two mitigations can improve the transparency of internalized knowledge. First, our agent architecture explicitly generates natural language reasoning traces (inner monologues) at each step, providing a degree of built-in interpretability even when the underlying knowledge is internalized. Second, we recommend that practitioners maintain comprehensive logs of all agent trajectories — including these reasoning traces, intermediate actions, and observations — to support diagnosis of any unintended behaviors that may have been inadvertently internalized during training.

### Acknowledgments

This work was supported by the Research Council of Finland (Flagship programme: Finnish Center for Artificial Intelligence FCAI, and grants 352986, 358246) and EU (H2020 grant 101016775 and NextGenerationEU). We acknowledge CSC for awarding this project access to the LUMI supercomputer, owned by the EuroHPC Joint Undertaking, hosted by CSC (Finland) and the LUMI consortium through Finland.

We would also like to thank Kalle Kujanpää for his contributions to the code, valuable insights, and support with the compute infrastructure, and Nicola Dainese for feedback on versions of this manuscript.

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

# A   Limitations

**Assumptions on task distribution**   A few assumptions should hold for the transfer from training tasks to test tasks to be most effective. First, we note that our training includes hints for meta-skills that help the agent in almost any task, such as using provided tools instead of inventing new ones, printing and verifying intermediate outputs, and effective and timely use of the `complete_task` call. Beyond these, tool usage should have some overlap between task instances (individual questions) for useful transfer: 1) the number of unique tools should grow slower than the number of task instances, and 2) for training tasks, some information about which tools are not needed should be available in order to improve the teacher prompts at training data generation time.

Although ToolQA divides question instances into task groups (which is one, but not the only way to define 2.) and furthermore some groups have overlap in their tool usage, these are not requirements for our method. On OfficeBench, our training process also leverages information about which apps are used in each task to narrow down the list of tool documentations to include in the teacher prompt. However, this varies on a per-task basis on OfficeBench.

In our experiments, we assume the agents' outputs can be automatically checked against the training tasks' ground truth answers to reduce human reviewing effort. If correct answers are not available, failures could be detected through a combination of a) trajectories not terminating correctly within the token budget b) errors raised while executing the agent's code actions and c) manual labeling.

**Human effort**   The proposed training pipeline depends on human supervision in the form of hints, which can be difficult to scale. However, as alluded to in Section 5.5, several measures are taken to reduce the amount of human effort: i) *only reviewing failed trajectories:* If correct answers are available for training tasks (as they are in ToolQA and OfficeBench), failure trajectories can be automatically detected. The human then only needs to review a subset of these to prepare training data for Rounds 2 and 3 (a subset because mistake patterns often repeat, but only need to be addressed once). ii) *reuse of hints in similar situations* with the help of automated reviewers: As soon as one mistake type is described, it is automatically labeled everywhere, so the human will need to review only a subset of failures.

With these optimizations, the total annotation effort for training our Round 2 and 3 agents was approximately 8 hours for ToolQA and 10 hours for OfficeBench, respectively. A breakdown of the time taken by each step of our annotation procedure is reported in Appendix D.4.

On the other hand, many alternative methods require comparable or even a higher amount of human effort: supervised fine-tuning would require full agent trajectories to be written, and reinforcement learning from human feedback would be much less sample-efficient in learning to generate good solutions than is possible with our targeted hints. Moreover, designing the initial hints in Round 1 corresponds to prompt engineering, a step required in any LLM application.

Reducing human oversight further would be a relevant direction for future work. Two concrete possibilities include: i) LLM-as-annotator, where a larger / more capable LLM replaces the human review step by identifying failure patterns and writing corrective hints automatically; and ii) environment-driven self-critique, where an agent self-corrects based on rich environmental feedback such as code execution errors, failed API calls, or automated task verification. Both approaches rely on preconditions that do not always hold – the first requires access to a sufficiently capable external model, and the second requires an environment that generates informative failure signals. Our method is designed to remain applicable in the more general case where neither condition is guaranteed.

# B  Additional results

## B.1  Coding and reasoning benchmarks

Table 6: Performance of the backbone LLMs on coding and reasoning benchmarks. The MNM results correspond to agents from Table 2.

| Agent | Trained on | HumanEval pass@1 | GSM8K |
|-------|-----------|------------------|-------|
| Llama-3.1-70B | - | $80.3_{0.2}$ | $92.8_{0.3}$ |
| MNM round 1 |  | $81.3_{0.2}$ | $92.8_{0.1}$ |
| MNM round 2 | ToolQA | $82.1_{0.2}$ | $92.8_{0.1}$ |
| MNM round 3 |  | $81.9_{0.4}$ | $92.6_{0.3}$ |
| MNM round 1 |  | $81.5_{0.3}$ | $92.1_{0.3}$ |
| MNM round 2 | OfficeBench | $81.3_{0.2}$ | $93.1_{0.1}$ |
| MNM round 3 |  | $80.7_{0.4}$ | $92.4_{0.1}$ |

## B.2  Cross-entropy loss

Table 7: ToolQA success rates for two variants of the distillation procedure. KL training on all trajectories refers to the same MNM agents as reported in Table 2.

| Loss | Trajectories | Round | DBLP | Agenda | Yelp | Airbnb | Flight | Coffee | Avg. |
|------|-------------|-------|------|--------|------|--------|--------|--------|------|
| KL | all | 1 | $92.1_{2.0}$ | $94.4_{2.1}$ | $84.7_{1.2}$ | $89.5_{2.0}$ | $92.2_{1.5}$ | $97.7_{0.9}$ | $91.8_{0.7}$ |
| KL | all | 2 | $97.7_{1.1}$ | $92.9_{1.4}$ | $94.9_{1.0}$ | $\mathbf{96.1}_{1.1}$ | $93.3_{1.0}$ | $\mathbf{99.5}_{0.5}$ | $95.7_{0.4}$ |
| KL | all | 3 | $\mathbf{98.9}_{0.6}$ | $96.0_{0.8}$ | $\mathbf{98.9}_{1.1}$ | $\mathbf{96.1}_{0.0}$ | $\mathbf{98.3}_{0.1}$ | $\mathbf{99.5}_{0.5}$ | $\mathbf{97.9}_{0.3}$ |
| cross-entropy | success only | 1 | $94.9_{0.0}$ | $\mathbf{99.2}_{0.8}$ | $84.2_{0.6}$ | $88.2_{0.0}$ | $91.1_{0.6}$ | $95.8_{0.0}$ | $92.2_{0.2}$ |
| cross-entropy | success only | 2 | $97.7_{0.6}$ | $96.8_{0.8}$ | $88.7_{1.1}$ | $86.9_{2.4}$ | $93.3_{1.0}$ | $98.6_{0.0}$ | $93.7_{0.5}$ |
| cross-entropy | success only | 3 | $97.7_{0.6}$ | $97.6_{0.0}$ | $97.7_{0.6}$ | $93.5_{0.7}$ | $96.1_{1.1}$ | $97.7_{0.5}$ | $96.7_{0.3}$ |

## B.3  Ablations

We experimentally verify the contributions of both probabilistic hint dropout and the data balancing strategy (which is described in Appendix G). As shown in Tables 8a and 8b, a dropout rate of 0.9 yields the highest average success rates on both benchmarks, compared to alternative values 0.5, 0.7 and 1.0. We attribute this to several benefits of probabilistic dropout: i) it encourages the model to continue paying attention to new information in the prompt, ii) randomization of the dropped-out hints results in the model seeing different training samples in each epoch, effectively serving as data augmentation, and iii) internalizing all guidance at once is more difficult for the model, making it possibly resort to memorizing spurious correlations, which might be alleviated by focusing on a subset of hints at a time.

Similarly, data balancing brings a statistically significant improvement to model success rates. In our ablations in Tables 8c and 8d, it increases average success rates by 1.4 percentage points on ToolQA, and 1.5 on OfficeBench.

Table 8a: Ablation over hint dropout rate $p$ on ToolQA.

| Agent | Dropout rate | DBLP | Agenda | Yelp | Airbnb | Flight | Coffee | Avg. |
|---|---|---|---|---|---|---|---|---|
| MNM round 1 | 0.5 | $89.8_{2.6}$ | $77.8_{2.9}$ | $\mathbf{87.6}_{1.5}$ | $86.9_{1.7}$ | $83.9_{3.1}$ | $92.5_{1.2}$ | $86.4_{0.9}$ |
| | 0.7 | $93.2_{2.0}$ | $87.3_{2.9}$ | $81.9_{1.5}$ | $87.6_{2.4}$ | $87.2_{0.6}$ | $94.8_{0.9}$ | $88.7_{0.8}$ |
| | 0.9 | $92.1_{2.0}$ | $\mathbf{94.4}_{2.1}$ | $84.7_{1.2}$ | $89.5_{2.0}$ | $\mathbf{92.2}_{1.5}$ | $97.7_{0.9}$ | $\mathbf{91.8}_{0.7}$ |
| | 1.0 | $\mathbf{95.5}_{1.5}$ | $84.9_{2.9}$ | $84.7_{3.0}$ | $\mathbf{91.5}_{0.6}$ | $\mathbf{92.2}_{2.4}$ | $\mathbf{98.6}_{0.8}$ | $91.3_{0.8}$ |

Table 8b: Ablation over hint dropout rate $p$ on OfficeBench.

| Agent | Dropout rate | 1 app | 2 apps | 3 apps | Avg. |
|---|---|---|---|---|---|
| MNM round 1 | 0.5 | $76.3_{3.9}$ | $86.4_{2.5}$ | $69.9_{2.1}$ | $77.2_{2.5}$ |
| | 0.7 | $78.5_{1.1}$ | $79.0_{4.5}$ | $72.0_{7.1}$ | $76.4_{1.7}$ |
| | 0.9 | $\mathbf{81.7}_{3.9}$ | $86.4_{1.2}$ | $\mathbf{80.6}_{6.7}$ | $\mathbf{82.8}_{3.2}$ |
| | 1.0 | $80.8_{3.2}$ | $\mathbf{88.9}_{2.1}$ | $69.9_{2.2}$ | $79.4_{0.8}$ |

Table 8c: Ablation over data balancing on ToolQA.

| Agent | Data balancing | DBLP | Agenda | Yelp | Airbnb | Flight | Coffee | Avg. |
|---|---|---|---|---|---|---|---|---|
| MNM round 3 | ✓ | $\mathbf{98.9}_{0.6}$ | $\mathbf{96.0}_{0.8}$ | $\mathbf{98.9}_{1.1}$ | $\mathbf{96.1}_{0.0}$ | $98.3_{0.1}$ | $99.5_{0.5}$ | $\mathbf{97.9}_{0.3}$ |
| | ✗ | $96.1_{0.0}$ | $92.1_{0.8}$ | $97.2_{0.6}$ | $93.5_{0.7}$ | $\mathbf{100.0}_{0.0}$ | $\mathbf{100.0}_{0.0}$ | $96.5_{0.2}$ |

Table 8d: Ablation over data balancing on OfficeBench.

| Agent | Data balancing | 1 app | 2 apps | 3 apps | Avg. |
|---|---|---|---|---|---|
| MNM round 2 | ✓ | $87.1_{1.7}$ | $\mathbf{91.4}_{2.5}$ | $\mathbf{83.9}_{1.9}$ | $\mathbf{87.3}_{1.0}$ |
| | ✗ | $\mathbf{89.2}_{2.8}$ | $84.0_{2.5}$ | $\mathbf{83.9}_{3.2}$ | $85.8_{1.0}$ |
| MNM round 3 | ✓ | $89.2_{2.1}$ | $\mathbf{92.6}_{2.1}$ | $\mathbf{89.2}_{1.1}$ | $\mathbf{90.3}_{0.7}$ |
| | ✗ | $\mathbf{92.5}_{1.1}$ | $90.1_{3.3}$ | $83.9_{3.2}$ | $88.8_{1.3}$ |

## B.4 Cross-domain transfer

To assess cross-domain generalization, we conducted a held-out experiment in which MNM agents are trained on two ToolQA task groups and evaluated zero-shot on another two. We focus on the four tabular database task groups (Yelp, Airbnb, Flight, Coffee), which share the same underlying tool set.

Table 9: Cross-domain transfer within ToolQA tabular task groups. Each row corresponds to an MNM agent trained on two source domains and evaluated zero-shot on the two held-out domains (indicated by the column headers). Dashes indicate that the evaluation domain was included in training. The final row shows the single-task agent with task-specific tool hints as a reference baseline. No ToolQA training corresponds to the Single-task agent + task-specific tool hints from Table 1.

| Training domains | Yelp | Airbnb | Flight | Coffee |
|---|---|---|---|---|
| Yelp+Airbnb | - | - | 45.5 | 87.3 |
| Airbnb+Flight | 63.8 | - | - | 88.3 |
| Flight+Coffee | 64.4 | 69.9 | - | - |
| Yelp+Flight | - | 82.4 | - | 87.3 |
| Yelp+Coffee | - | 77.8 | 42.2 | - |
| Airbnb+Coffee | 63.3 | - | 50.6 | - |
| **Average** | 63.8 | **76.7** | 46.1 | **87.6** |
| **No ToolQA training** | **64.4** | 50.3 | **49.4** | 85.4 |

The results in Table 9 show that hints distilled for one pair of domains transfer positively to held-out domains in the case of Airbnb and Coffee, with a significant +26.4% to Airbnb. This suggests that the internalized skills, such as correct database loading, filtering, and value extraction, are genuinely transferable across tabular tasks.

Transfer to Flight is more challenging, likely reflecting that flight-specific reasoning (such as parsing flight identifiers and handling delay calculations) requires targeted experience that is harder to acquire from other tabular domains. Overall, these results support the generalizability of our approach within a shared tool ecosystem.

## B.5 Sensitivity to hint phrasing

To evaluate our method's robustness to hint variations, we conducted an experiment in which we asked Llama3.1-70B to rephrase the human-written hints while preserving semantic meaning. Results for ToolQA Round 1 are shown in Table 10. We observe the average success rate across ToolQA domains to be nearly identical with rephrased hints.

As an example of the applied rephrasing, below are a few of the rephrased hints:

- Before ending the task, print the final answer to check the value and the format of it. → Prior to completing the task, display the final answer to verify both its value and formatting.

- When a large number of documents are retrieved, it may not be feasible to display all of them at once, as the output could be truncated. → If you retrieve a very large set of documents, displaying them all at once may be impractical since the output is likely to be cut off.

- At each step, use only one tool, and carefully observe the output of each step to better understand how the tool functions. → Invoke just one tool per step, and pay close attention to each tool's output so you develop a clear understanding of how it works.

Table 10: ToolQA Round 1 performance using original versus rephrased hints.

| | DBLP | Agenda | Yelp | Airbnb | Flight | Coffee | Avg. |
|---|---|---|---|---|---|---|---|
| Original hints | 92.1 | 94.4 | 84.7 | 89.5 | 92.2 | 97.7 | 91.8 |
| Rephrased hints | 93.2 | 92.1 | 85.9 | 91.5 | 91.7 | 97.2 | 91.9 |

## C  Inference efficiency

MNM Round 3 agents achieve a 3-4x inference speedup relative to the base Llama-3.1-70B agent with combined hints. We calculated this speedup based on end-to-end task execution time, which includes sequential steps of agent execution until task completion. We observed an average runtime of approximately 90 seconds per ToolQA task for the MNM Round 3 agent, compared to 340 seconds for the untrained Llama-3.1-70B with combined hints, when running a vLLM server on 4 AMD MI250X GPUs. The equivalent numbers for OfficeBench are 140 seconds for our Round 3 agent vs. 600 seconds for Llama-3.1-70B.

The exact durations will depend on hardware, but using these numbers as a reference: assuming inference-time savings of 250 seconds per task and a total training time of 19.75 hours (16 + 1.25 + 2.5h, Appendix D.3) for ToolQA on identical hardware, the break-even point is 285 sequential task executions, a threshold readily exceeded in any realistic deployment scenario. For OfficeBench, assuming 460 seconds saved per task, the break-even point is even earlier: 32.5 hours (27 + 3.5 + 2h) of training is compensated for by the inference savings for 255 sequential task executions. Note that these calculations assume sequential task execution and are therefore conservative: in practice, batched inference allows multiple tasks to be processed in parallel, reducing the wall-clock break-even time proportionally.

Table 11: The average number of tokens used to solve test tasks.

| Agent | Input tokens, ToolQA | Output tokens, ToolQA | Input tokens, OfficeBench | Output tokens, OfficeBench |
|---|---|---|---|---|
| DeepSeek-V3 | 59,056 | 552 | 58,728 | 931 |
| GPT-4o | 77,736 | 527 | 60,951 | 776 |
| Llama-3.1-70B | 74,950 | 449 | 72,541 | 604 |
| MNM round 3 (ours) | 5,564 | 460 | 9,933 | 630 |

## D  Training details

In all training rounds, we use the AdamW optimizer (Loshchilov, 2017) with a maximum learning rate of $10^{-5}$, a linear warm-up over 30 update steps and batch size 8. The optimizer, learning rate, the number of training epochs in each round and other hyperparameters were chosen based on validation loss.

The student model in each round is constructed by augmenting the base Llama model with a full precision LoRA adapter of rank 128, applied to all (linear) layers. After training, the adapter is merged to the base model weights and a new LoRA of the same rank is added in each subsequent round. The rank was chosen as the largest that could be trained on a single compute node (with 4 AMD MI250X GPUs) in our cluster architecture.

For ToolQA, we collect three teacher trajectories per task in every round. For OfficeBench, four teacher trajectories per task are collected in the first round, followed by three trajectories per task in subsequent rounds.

### D.1  ToolQA

**Round 1:** By solving each training task three times, we compile 7,107 state-action-hint triplets, which form the training set $\mathcal{D}_1 = \{(s, a, h_1(s)\}$. We then train the student model for 2 epochs, where each update involves randomly sampling a batch from the training set.

**Round 2:** We collect a total of 208 state-action-hint triplets, derived from 99 unique states where multiple actions were sampled per state. Of the 208 collected state-action-hint triplets, 184 samples are used to form the training set $\mathcal{D}_2 = \{(s, a, h_2(s))\}$, while 24 samples are reserved for tracking validation loss during training. To refine the agent further, we construct a new student model by adding another LoRA adapter to the latest trained model and train for 6 epochs, using randomly sampled batches from $\mathcal{D}_2$.

**Round 3:** By sampling multiple actions from the 68 detected states, we create 120 state-action-hint triplets. However, this dataset poses two potential challenges: 1) the relatively small number of unique states, as the agent had already improved significantly; 2) an imbalance in task coverage, since the agent performs some task types more accurately than others.

To prevent performance degradation on task types underrepresented in this dataset, we also collect full trajectories using the trained agent with the hints designed in Round 1. We create a training set $\mathcal{D}_3$ by combining the 120 corrected transitions with 69 new full trajectories, ensuring that all task types were well represented. The final dataset used in this round comprises 359 training samples.

We then add a new adapter and again train the updated model for 6 epochs.

## D.2 OfficeBench

**Round 1:** We compile 3,415 state-action-hint triplets and train the student model for 7 epochs.

**Round 2:** A total of 198 state-action-hint triplets are derived from 66 unique states with mistakes. Of the 198 collected state-action-hint triplets, 178 samples are used to form the training set, while 20 samples are use to for the validation set.

To create a more balanced dataset, we augment this with 473 states from 58 new full trajectories. The agent is trained for 6 epochs on this balanced dataset consisting of 651 training samples.

**Round 3:** We obtain 89 state-action-hint triplets from 30 unique detected states, where 80 triplets are used for training and 9 triplets are used to form the validation set.

As per our balancing strategy, we add 44 new full trajectories (278 samples) to the training set. The agent is finally trained for 6 epochs on 358 training samples.

## D.3 Computational cost

The training time per round was 16h, 1.25h, and 2.5h for ToolQA, respectively, and 27h, 3.5h and 2h for OfficeBench. All training rounds were run on four AMD MI250X GPUs (a total of 8 GPU chips).

## D.4 Human annotation cost

We also report the estimated amount of human expert annotation time for training our models. The observed time to arrive at a finalized hint is approximately 19 minutes each (a breakdown per subtask is shown in Table 12), assuming prior familiarity with the task domain. This calculation includes any required iterations of reviewer prompt refinement and hint refinement. On average, 0.5 prompt refinement and 0.5 hint refinement iterations were needed for each identified mistake type before the outputs were deemed of sufficient quality (in both the accuracy of filtering and the automatically-evaluated mistake correction rate).

This gives a total annotation time of 7.6 hours for 24 hints on OfficeBench, and 10.1 hours for 32 hints on ToolQA. This calculation assumes an efficient annotation pipeline in which LLM inference can happen asynchronously from human annotation time.

Table 12: Estimated human annotation time per finalized hint.

| Annotation task | Time (min) |
|---|---|
| 1. Mistake identification (reading a failed trajectory) | 5 |
| 2. Reviewer prompt writing | 2 |
| 3. Reviewer prompt verification (inspecting a subset of filtered steps) | 3 |
| 4. Reviewer prompt refinement (if needed) | 0.5x1 |
| 5. Refined reviewer prompt verification (if needed) | 0.5x3 |
| 6. Hint writing | 2 |
| 7. Hint verification (inspecting a subset of corrected steps) | 3 |
| 8. Hint refinement (if needed) | 0.5x1 |
| 9. Refined hint verification (if needed) | 0.5x3 |
| **Total per finalized hint** | **19** |

## E   Reviewer evaluation

We individually evaluate the accuracy of our automated mistake-localization pipeline across all hint types in both rounds of corrective training. For each filter, we report the false positive rate (FPR; fraction of correct trajectories incorrectly flagged as containing a mistake) and the false negative rate (FNR; fraction of mistake-containing trajectories missed by the filter). The results for ToolQA are included in Tables 13 and 14, and for OfficeBench in Table 15.

Table 13: False positive (FPR) and negative rates (FNR) of LLM-based and script-based mistake filters for ToolQA Round 2. Script-based filters are only applicable to a subset of mistake types; dashes indicate that no script-based filter could be designed for that mistake type.

| Mistake type | LLM FPR % | LLM FNR % | Script FPR % | Script FNR % |
|---|---|---|---|---|
| Tool parameters | 0.4 | 0.1 | - | - |
| Time window | 0.5 | 1.3 | - | - |
| Coffee price | 0.1 | 0.2 | - | - |
| Yelp appointment | 0 | 0 | 0 | 0 |
| Yelp categories | 0.2 | 0.3 | 0 | 0 |
| Duplicate cities | 0.4 | 0.6 | - | - |
| Radius | 0.1 | 0 | - | - |
| Column name | 0 | 0 | 0 | 0 |
| Load database | 0 | 0 | 0 | 0 |
| Price value | 0.8 | 0 | - | - |
| Review rate | 0.6 | 1.0 | - | - |
| Distance | 0.2 | 0 | - | - |
| Delay minutes | 0.4 | 0 | 0 | 0 |
| Incorrect time format | 0.6 | 0 | - | - |
| Average | 0.3 | 0.3 | - | - |

Table 14: False positive (FPR) and negative rates (FNR) of LLM-based and script-based mistake filters for ToolQA Round 3. Script-based filters are applicable to only 2 of the 20 mistake types.

| Mistake type | LLM FPR % | LLM FNR % | Script FPR % | Script FNR % |
|---|---|---|---|---|
| Date missing | 0 | 0 | - | - |
| Format conversion | 0.3 | 2.5 | - | - |
| Id value | 0 | 0 | - | - |
| Recommendation | 0.2 | 1.6 | - | - |
| Review number | 0 | 0 | - | - |
| Arrdelay printing | 0 | 0 | - | - |
| Average time nan | 0.1 | 0 | - | - |
| Flights diverted | 0.2 | 0 | - | - |
| Flights ids | 0 | 1.3 | - | - |
| Speed nan value | 0 | 0 | - | - |
| Taxi in printing | 0 | 0 | - | - |
| Citations | 0.4 | 0 | - | - |
| Duplicate citations | 0.5 | 2.2 | - | - |
| Graph loading | 0 | 0 | 0 | 0 |
| Graph name | 0 | 0 | - | - |
| Text printing | 0 | 0 | - | - |
| Text hard question solution | 0 | 3.7 | - | - |
| Time window | 0.4 | 1.2 | - | - |
| Yelp appointment | 0 | 0 | - | - |
| Yelp categories | 0 | 0 | 0 | 0 |
| Average | 0.1 | 0.7 | - | - |

Table 15: Average False positive (FPR) and negative rates (FNR) of LLM-based mistake filters for OfficeBench. No script-based filters were applicable for OfficeBench mistake types.

| Round | Average FPR % | Average FNR % |
|---|---|---|
| Round 2 | 0.6 | 0.3 |
| Round 3 | 0.4 | 0 |

## F   Example tasks

Table 16: Example question templates and possible solution strategies from the ToolQA benchmark.

| Group | Question template | Solution steps |
|---|---|---|
| Yelp | Which [business category] has the highest review count in [city], [state]? | 1. Load Yelp database. 2. Filter by [city] and [state]. 3. Select rows for which the category column includes [business category]. 4. Fetch the review count column. 5. Use the index of the max review count to select and return the name of the business. |
| Flights | How many extra minutes did the [flight ID] flight take from [departure] to [destination] on [date]? | 1. Load flights database. 2. Separate [flight ID] into [airline ID] and [flight number]. 3. Filter by [airline ID], [flight number], [departure], [destination] and [date]. 4. Fetch departure delay and arrival delay columns and return arrival delay - departure delay. |

Table 17: Example tasks and possible solution strategies from the OfficeBench benchmark.

| Task description | Apps used | Solution steps |
|---|---|---|
| Company increases base by 5% / yr, stock by 10% / yr, bonus by 10% / yr, show this in salary excel file with new header *amount after 1 year.* Round down with no decimal. | Excel | 1. List files in data_path to identify the salary excel file. 2. Read salary.xlsx. 3. Multiply the salary components by the specified percentages using Python and round down to the nearest integer. 4. Add the numbers in a new column in salary.xlsx, in the order indicated by the row headers, and give the new column the header *amount after 1 year.* |
| Convert meeting agenda from calendar meeting_agenda into Excel file meeting_agenda.xlsx for further analysis, email it to manager Rena. | Calendar, Excel, Email | 1. List events in calendar meeting_agenda. 2. Create a new Excel file meeting_agenda.xlsx in data_path. 3. For each event, write its title, start and end times, description and location into a new row in the spreadsheet. 4. Write an email including the same information and send it to Rena. |

## G   Data balancing strategy

In each round $i$ (where $i = 2, \ldots, I$), we apply a data balancing strategy to ensure that the training data for round $i$ includes a more even distribution of states across different task groups and different types of failure cases. To prepare the data for balancing, in addition to running the agent on the training set without hints (to collect failure cases), we also run the agent with hints $h_1$ to generate an auxiliary set of candidate trajectories, denoted as $D'_i$.

The balancing procedure is applied in three sequential steps.

**1. Insufficient mistakes:**

For some tasks $T$, the number of collected states with specific mistakes may be too small (e.g., only a single state is available) to provide sufficient and diverse data for training.

- **ToolQA example:** In round 3, only one state was collected for the failure cases "incorrectly interpret flight IDs" and "incorrectly view the attribute."

- **OfficeBench example:** In round 2, only one state was collected for "using an incorrect Excel file path." In round 3, only one state was collected for each of the failure cases "incorrectly reading an Excel file," "incorrectly writing content into a Word file," and "incorrectly using the OCR tool."

**Balancing action:** For ToolQA, we randomly sample **one** additional trajectory from $D_i'$ that shares the same question template as $T$. For OfficeBench, where tasks are not explicitly tied to templates, we instead sample **one** additional trajectory from $D_i'$ that involves the same applications as $T$.

The selected trajectories are added to the training data for round $i$.

**2. Imbalanced representation across task groups and involved applications:**

We again examine the distribution of states in the training data for round $i$:

**ToolQA:** We sample additional trajectories from $D_i'$ for tasks in underrepresented groups (e.g., *Coffee*), as well as for tasks in groups with a higher number of failure cases observed in previous rounds (e.g., *Yelp*).

**OfficeBench:** We sample additional trajectories from $D_i'$ for tasks involving applications that are underrepresented in the collected states (e.g., *Calendar*), as well as for tasks involving applications with a higher number of failure cases identified in previous rounds (e.g., *Excel* and *Email*).

Again, the selected trajectories are added.

**3. Preserving learned skills:**

To prevent the loss of previously learned skills, we sample trajectories from tasks not yet covered in the collected states or in Steps 1 and 2. The number of such trajectories is determined by balancing the counts of states collected from agent failures vs. trajectories added through balancing.

## H   Tool documentation

The documentation of each tool consists of a one-line description of the tool, explanation of each input argument, a description of the function output, and 1-2 example calls.

## H.1 ToolQA tools

**Listing 2** Tool documentation for ToolQA-DBLP.

```
load_graph() -> tuple
        This function loads the graph data.
        - Output: The graph data.
        Example calls:
        'GRAPH_DATA = load_graph()': Returns the graph data.
-------
check_nodes(GRAPH_DATA: tuple, graph_type: str, node: str) -> dict
        This function returns the detailed attributes of a given node in the graph.
        When the graph is 'PaperNet', the node attributes include 'title', 'authors', 'year', 'venue', 'number of citations for this
        ↪ paper', 'keywords', 'doc_type', 'page_start', and 'page_end'.
        When the graph is 'AuthorNet', the node attribute is only the organization of the author, without any information on papers.
        - Input:
            1. Graph data as a tuple;
            2. The graph name, which can be either 'PaperNet' or 'AuthorNet'.
            3. The node name. When the graph is 'PaperNet', the node name must be the title of the paper. When the graph is
            ↪ 'AuthorNet', the node name must be the author's name.
        - Output: The attributes of the given node in the graph.
        Example calls:
        'check_nodes(GRAPH_DATA, 'PaperNet', 'Learning the Principle of Least Action with Reinforcement Learning.')': Returns
        ↪ {'title': 'Learning the Principle of Least Action with Reinforcement Learning.', 'authors': [{id:'', name: 'Hao Zhang',
        ↪ org: 'Univ Tokyo, Inst Ind Sci, Tokyo, Japan'},], 'year': 2021, 'venue': {'raw': 'AAAI Spring Symposium - MLPS'}, 'number
        ↪ of citations for this paper': 0, 'keywords': [], 'doc_type': 'Conference', 'page_start': '', 'page_end': ''}.
-------
check_neighbours(GRAPH_DATA: tuple, graph_type: str, node: str) -> list[str]
        This function returns the names of a given node's neighbours in the graph.
        When the graph is 'AuthorNet', the output is a list of author names who have co-authored with the given author, without any
        ↪ information on papers.
        - Input:
            1. Graph data as a tuple;
            2. The graph name, which can be either 'PaperNet' or 'AuthorNet'.
            3. The node name.
        - Output: A list of the names of a given node's neighbours in the graph.
        Example calls:
        'check_neighbours(GRAPH_DATA, 'AuthorNet', 'Chao Zhang')': Returns ['YUHUI YUAN', 'Rao Fu', 'Lang Huang'].
-------
check_edges(GRAPH_DATA: tuple, graph_type: str, node1: str, node2: str) -> dict
        This function returns the detailed attributes of an edge between two nodes in the graph.
        When the graph is 'AuthorNet', the edge represents the collaboration between two authors, and the edge attributes are
        ↪ information of the papers they have co-authored.
        - Input:
            1. Graph data as a tuple;
            2. The graph name, which can be either 'PaperNet' or 'AuthorNet'.
            3. The first node name.
            4. The second node name.
        - Output: The attributes of the edge between the two given nodes in the graph.
        Example calls:
        'check_edges(GRAPH_DATA, 'AuthorNet', 'Chao Zhang', 'Weihong Lin')': Returns {'weight': 1, 'papers': ['HRFormer:
        ↪ High-Resolution Vision Transformer for Dense Predict.'], 'number of citations for this paper': [95]}.
```

**Listing 3** Tool documentation for ToolQA Yelp, Airbnb, Flight and Coffee tasks.

```
load_db(db_variant: str) -> pandas.DataFrame
      This function loads the database and shows names of all columns of this database. 'db_variant' can be one of the following:
      ↪ flights/coffee/airbnb/yelp.
      - Input: used to indicate which database to load.
      - Output: A 'pandas.DataFrame' containing the loaded database. Note that all elements in the DataFrame are converted to
      ↪ strings.
      Example calls:
      'flights_db = load_db('flights')': Returns the flights database.
-------
data_filter(database: pandas.DataFrame, argument: str) -> pandas.DataFrame
      This function filters the database by the column name, the relation (e.g., =, >, etc.) and the value, and keeps only rows
      ↪ matching the argument.
      - Input:
         1. A database;
         2. An argument string with column names, relations (e.g., =, >, etc.) and values as conditions to filter the database.
         ↪ Conditions in the argument should be separated by a semicolon and a space '; '.
         Example Argument: 'name=Snip Philadelphia; postal_code=19130'
         Note: You must strictly follow the extact format of input of the argument.
      - Output: A new 'pandas.DataFrame' containing rows matching the argument.
      Example calls:
      'new_database=data_filter(full_database, 'IATA_Code_Marketing_Airline=AA; Flight_Number_Marketing_Airline=5647; Origin=BUF;
      ↪ Dest=PHL; FlightDate=2022-04-20')': Returns a 'new_database' contains only one row of the database.
-------
get_value(data: pandas.DataFrame, name: str) -> str
      This function returns the value(s) of a specific column in the database.
      - Input:
         1. A database;
         2. A single column name
         Note: You can only input one column name.
      - Output: The value(s) of the specific column as a string. If there are multiple rows in this database, the values are
      ↪ concatenated using a comma and a space ', '.
      Example calls:
      'dep_time = get_value(new_database, 'DepTime')': Returns '1752.0'.
      'flight_number = get_value(new_database, 'Flight_Number_Marketing_Airline')': Returns '388, 844, 3517, 4301'.
```

**Listing 4** Tool documentation for ToolQA-Agenda.

```
retrieve_agenda(query: str, num_docs: int) -> list[str]
   This function retrieves relevant agenda-related documents based on the query.
   - Input:
      1. query: A query including the key information to answer the question.
      2. num_docs: The number of documents to retrieve. Determine it based on the complexity of the question.
   - Output: A list of 'num_docs' documents relevant to the query.
   Example calls:
   'retrieved_docs = retrieve_agenda("Grace attend Broadway Show on February 2nd, 2022", 10)': Returns 10 revelant documents.
```

**Listing 5** Documentation for the `complete_task` tool in ToolQA (appended to tool documentation list in all tasks).

```
complete_task(final_report: str, return_value: str)
   Complete the task and report the final answer. It is recommended that you do this command as the only command in the cell.
   The return_value needs to be a string.
   Example calls:
      complete_task("The star rating of Perenn Bakery is", "4.5")
   or
      complete_task("The meeting should be scheduled at", "9:00 AM-6:00 PM")
```

## H.2 OfficeBench tools

**Listing 6** Documentation for the `complete_task` tool in OfficeBench.

```
complete_task(final_report: str, return_value=None)
   Complete the task. It is recommended that you do this command as the only command in the cell.
   If the task requires providing a final answer, the return_value needs to be a string.
   Example calls:
      complete_task("The event has been created to the calendar.") # This task does not require providing a final answer.
      complete_task("The email subject is:", "Meeting notes") # This task requires providing a final answer.
```

**Listing 7** Documentation for tools related to the Email application.

```
list_emails(username)
    List all emails in a user's inbox.
    - Input: username: the user name
    - Output: a string containing information for all emails.
    Example call:
        'list_emails('Sam')': Returns "Email ID: meeting.eml
From: Alice@emaildomain.com
To: Sam@example.com
Subject: Meeting
Content: Team meeting at 3pm...

-------
read_email(username, email_id)
    Read a user's email by the given Email ID.
    - Input:
        1. username: the user name
        2. email_id: the email's unique ID
    - Output: a string containing the email information.
    Example call:
        'read_email('Anna', 'reminder.eml')' Returns "From: Sam@example.com
To: Anna123@hotmail.com
Subject: Reminder
Content: Remember the due date tomorrow...
"

-------
send_email(sender, recipient, subject, content)
    Send an email from the sender to the recipient. The email's file name is {subject}.eml.
    - Input:
        1. sender: the sender's email address
        2. recipient: the recipient's email address
        3. subject: the email subject. The email's file name is the same as the subject.
        4. content: the email content
    - Output: "Successfully sent email to {recipient}." if the email is sent successfully, an error message otherwise.
    - Example call:
        'send_email('Alice@example.com', 'Bob@domain.com', 'Meeting', 'Hello, Bob!')': Returns 'Successfully sent email to Bob.'
```

**Listing 8** Documentation for OfficeBench tools related to the OCR (Optical Character Recognition) application.

```
ocr_recognize_text(image_path)
    Recognize text from an image using Optical Character Recognition (OCR).
    - Input: file_path: the path to the image file
    - Output: the recognized text from the image.
    Example call: 'ocr_recognize_text(os.path.join(data_path, 'image.jpg'))': Returns "ABC DEF"
```

**Listing 9** Documentation for OfficeBench tools related to the Calendar application.

```
create_event(user: str, summary: str, time_start: str, time_end: str)
    Create a new event to a user's calendar where the time format is '%Y-%m-%d %H:%M:%S'.
    - Input:
        1. user: the user name
        2. summary: the event summary exactly as mentioned in the task. You may include key details, but ensure it matches the
        ↪ original wording.
        3. time_start: event start time
        4. time_end: event end time
    - Output: True if the event is created successfully, False otherwise.
    Example call: 'create_event('Tom', 'meeting with the staff in Room 12 to discuss the project', '2024-05-17 10:00:00', '2024-05-17
    ↪ 11:00:00')': Returns True

-------
delete_event(user, summary)
    Delete an event from a user's calendar given the event summary.
    - Input:
        1. user: the user name
        2. summary: the event summary
    - Output: True if the event is deleted successfully, False otherwise.
    Example call: 'delete_event('Sarah', 'lunch')': Returns True

-------
list_events(username)
    List all events from a user's calendar.
    - Input: username: the user name
    - Output: a string containing information for all events.
    Example call:
        'list_events('Mike')': Returns "Summary: Meeting
Start Time: 2024-05-13 09:00:00
End Time: 2024-05-13 10:00:00
Description: Team meeting
Location: Office"
```

**Listing 10** Documentation for OfficeBench tools related to the PDF application.

```
pdf_read_file(file_path)
    Read the content of a PDF file.
    - Input: file_path: the path to the PDF file
    - Output: the text content of the PDF file.
    Example call: 'pdf_read_file(os.path.join(data_path, 'test.pdf'))': Returns "Test"

-------
image_convert_to_pdf(image_file_path, pdf_file_path)
    Convert an image file to a PDF file.
    - Input:
        1. image_file_path: the path to the image file
        2. pdf_file_path: the path to the PDF file
    - Output: True if the conversion is successful, False otherwise.
    Example call: 'image_convert_to_pdf(os.path.join(data_path, 'image.jpg'), os.path.join(data_path, 'test.pdf'))': Returns True

-------
pdf_convert_to_image(pdf_file_path, image_file_path)
    Convert a PDF file to an image file.
    - Input:
        1. pdf_file_path: the path to the PDF file
        2. image_file_path: the path to the image file
    - Output: True if the conversion is successful, False otherwise.
    Example call: 'pdf_convert_to_image(os.path.join(data_path, 'test.pdf'), os.path.join(data_path, 'image.jpg'))': Returns True

-------
pdf_convert_to_word(pdf_file_path, word_file_path)
    Convert a PDF file to a Word file.
    - Input:
        1. pdf_file_path: the path to the PDF file
        2. word_file_path: the path to the Word file
    - Output: True if the conversion is successful, False otherwise.
    Example call: 'pdf_convert_to_word(os.path.join(data_path, 'test.odf'), os.path.join(data_path, 'test.docx'))': Returns True
```

**Listing 11** Documentation for OfficeBench tools related to Excel.

```
excel_create_new_file(file_path)
    Create a new Excel file.
    - Input: file_path: the path to the new Excel file
    - Output: True if the file is created successfully, False otherwise.
    Example call: 'excel_create_new_file(os.path.join(data_path, 'test.xlsx'))': Returns True

-------
excel_delete_cell(file_path, row_idx, column_idx, sheet_name=None)
    Delete the content of a cell in an Excel file.
    - Input:
        1. file_path: the path to the Excel file
        2. row_idx: the row index of the cell
        3. column_idx: the column index of the cell
        4. sheet_name: the name of the sheet (optional). If not provided, the active sheet will be used.
    - Output: True if the cell content is deleted successfully, False otherwise.
    Example call: 'excel_delete_cell(os.path.join(data_path, 'test.xlsx'), 2, 1)': Returns True

-------
excel_read_file(file_path, sheet_name=None)
    Read the content of an Excel file.
    - Input:
        1. file_path: the path to the Excel file
        2. sheet_name: the name of the sheet (optional). If not provided, the active sheet will be used.
    - Output: a string containing the spreadsheet entries in "({row_idx}, {col_idx}): {value}" format.
    Example call:
        'excel_read_file(os.path.join(data_path, 'test.xlsx'))': Returns "(1, 1): Order 1 (1, 2): 123
(2, 1): Order 2 (2, 2): 456
"

-------
excel_set_cell(file_path, text, row_idx, column_idx, sheet_name=None)
    Write text to a cell in an Excel file.
    - Input:
        1. file_path: the path to the Excel file
        2. text: the text to write, which must be a string.
        3. row_idx: the row index of the cell
        4. column_idx: the column index of the cell
        5. sheet_name: the name of the sheet (optional). If not provided, the active sheet will be used.
    - Output: True if the text is written to the cell successfully, False otherwise.
    Example call: 'excel_set_cell(os.path.join(data_path, 'test.xlsx'), 'Hello, World!', 1, 1)': Returns True

-------
excel_convert_to_pdf(excel_file_path, pdf_file_path)
    Convert an Excel file to a PDF file.
    - Input:
        1. excel_file_path: the path to the Excel file
        2. pdf_file_path: the path to the PDF file
    - Output: True if the conversion is successful, False otherwise.
    Example call: 'excel_convert_to_pdf(os.path.join(data_path, 'test.xlsx'), os.path.join(data_path, 'test.pdf'))': Returns True
```

**Listing 12** Documentation for OfficeBench tools related to Microsoft Word (1 / 2).

```
word_create_new_file(file_path)
    Create a new Word file.
    - Input: file_path: the path to the new Word file
    - Output: True if the file is created successfully, False otherwise.
    Example call: 'word_create_new_file(os.path.join(data_path, 'test.docx'))': Returns True

-------
word_read_file(file_path)
    Read the content of a Word file.
    - Input: file_path: the path to the Word file
    - Output: the text content of the Word file.
    Example call: 'word_read_file(os.path.join(data_path, 'test.docx'))': Returns "Document contents here."
```

**Listing 12** Documentation for OfficeBench tools related to Microsoft Word (2 / 2).

```
word_write_to_file(file_path, contents)
    Write text to a Word file.
    - Input:
        1. file_path: the path to the Word file
        2. text: the text to write
    - Output: True if the text is written to the file successfully, False otherwise.
    Example call: 'word_write_to_file(os.path.join(data_path, 'test.docx'), 'New content')': Returns True

-------
word_convert_to_pdf(word_file_path, pdf_file_path)
    Convert a Word file to a PDF file.
    - Input:
        1. word_file_path: the path to the Word file
        2. pdf_file_path: the path to the PDF file
    - Output: True if the conversion is successful, False otherwise.
    Example call: 'word_convert_to_pdf(os.path.join(data_path, 'test.docx'), os.path.join(data_path, 'test.pdf'))': Returns True
```

# I   Data preprocessing

## I.1   ToolQA

### I.1.1   Task split

We evaluate ToolQA under two split schemes.

- **Default split (all tables except Table 5):** training, validation and test sets are *split by task*, so each question template appears in all splits.

- **Unseen-template split (Table 5):** the splits are *by template*, so the agent is evaluated on held-out templates.

For examples of task templates, see Table 16 in Appendix F. Several tasks are created from a template by filling in entity placeholders.

### I.1.2   Task selection

The ToolQA benchmark consists of eight task groups. However, we focus on six of them in this work and exclude tasks from the GSM8K and SciREX groups for the following reasons: (1) Since our agent can execute Python code for mathematical calculations, it does not require additional tools for arithmetic reasoning. As a result, training the agent to use the tools provided in ToolQA for the GSM8K group is unnecessary; (2) Similar to the Agenda tasks, SciREX requires the agent to write queries for retrieving relevant documents from an external corpus stored in a Chroma database. This retrieval relies on measuring embedding similarity between queries and documents. However, we find this mechanism to be unreliable, where for most of tasks in this group, relevant documents are often not consistently retrieved across multiple human-written queries. Due to this limitation in the document retrieval step, we could not use the SciREX tasks to fairly and accurately evaluate agent performance. In addition, since Agenda tasks already assess the agent's ability to retrieve documents by generating effective queries, we consider SciREX redundant and exclude it in this work.

For tasks in the chosen task groups, we find issues related to information mismatch and missing data. To ensure the solvability of tasks, we applied the following processing and filtering steps to exclude any flawed or unsolvable instances:

**DBLP:** We find inconsistencies in the original index files `id2author_dict` vs. `author2id_dict` and `id2title_dict` vs. `title2id_dict`, likely caused by how duplicate author and paper names were handled during dictionary creation. In addition, there are mismatches between the DBLP graph data and the dictionaries connecting authors and papers. To address these issues, we: (1) remove mismatched author-id and paper-id entries from the index files; and (2) manually review each task and exclude those where the correct answers could not be reliably

found by any agent. To ensure a sufficient number of tasks, we also generated additional tasks following the official dataset generation process. After processing, we retain 168 tasks across 15 question templates.

**Agenda:** Some tasks in this group show similar issues to those in SciREX, where relevant documents were not consistently retrievable, making the correct answers inaccessible. After manually reviewing each task, we keep only those that are solvable in the agent framework, resulting in 140 tasks from 7 question templates.

**Yelp:** We find the following issues in this task group: (1) Incomplete task descriptions missing key information (e.g., address); (2) Tasks matching multiple rows in the tabular database, leading to non-unique or unclear correct answers (e.g., in tasks involving finding the nearest business to a given address). These tasks were removed to ensure reliable evaluation. Additionally, we discovered errors in the original dataset generation process, particularly in distance calculations. After correcting these mistakes, the correct answers were updated accordingly. After processing, 180 tasks from 20 question templates are retained.

**Airbnb:** Similar to the Yelp tasks, this task group shows issues with multiple matching rows in the database. Additionally, some tasks contained incorrect ground truth answers. We address these issues by removing tasks with ambiguous answers and correcting errors in the provided answers where necessary. We keep 168 tasks across 17 question templates.

**Flight:** Tasks under the original template *How many flights have a distance greater than 500 miles on* `[flight date]` do not align with the official dataset generation process, as the answers are actually generated based on a 300-mile threshold rather than 500 miles. We update the task description to reflect the 300-mile threshold. After verification, we retain all tasks from the original dataset, resulting in 200 tasks across 20 question templates.

**Coffee:** We update the task descriptions and answers for several Coffee tasks to address the following issues: (1) For tasks with the question template *What was the coffee price range from* `[date 1]` *to* `[date 2]?`, the answers were generated incorrectly. We corrected the answers to align with the specific questions. Additionally, we revised the task description to *What was the difference between the highest and the lowest coffee price from* `[date 1]` *to* `[date 2]?` to eliminate any ambiguity. (2) For tasks using the template *What was the percentage change in coffee price on* `[date]` *compared to the previous day?*, some entries lack data for the *previous day* in the database. We update these tasks to ensure they are solvable. As a result, 230 tasks from 21 question templates remain.

## I.2  OfficeBench

OfficeBench consists of a total of 300 tasks, each requiring the use of 1 to 3 applications to complete.

During our review, we identified several issues across the tasks, including: errors in task files, incorrect evaluation conditions, mismatches between task descriptions and expected answers, and incomplete or ambiguous task descriptions. Additionally, some errors arose from limitations of the provided tools. For example, the OCR tool sometimes misidentifies text in images.

Tasks that could not be corrected through modifications to their descriptions or evaluation conditions, such as those with faulty source files or tool-related problems, were removed. After filtering, 286 tasks remained. For tasks with fixable issues, such as vague descriptions (e.g., missing format requirements that are expected in the answer), we revised them to ensure clarity and solvability, and adjusted the evaluation conditions to ensure fairness.

After task processing, we randomly split these tasks into training, validation, and test sets with a ratio of 67:3:30, respectively.

## J   Example agent prompt

---

**Listing 13** Prompt for an example ToolQA DBLP task at time $t = 0$ with task-specific guidance.

```
<task_description>
In this task, you are asked to answer a question by retrieving information from an external database. You can use provided tools to
↪  assist in exploring the database.

Here is the question:
Who are the authors of "Perfectly Clustering Words are Primitive Positive Elements of the Free Group."? Seperate the authors with
↪  comma and space ', '.

When you complete the task, please instantiate and return an object of class str.
</task_description>

Project started at 2025-01-10 20:10:23

<guidelines>
Response format

Please use the following format to structure your inner monologue and tool calls:
<inner_monologue>
Your thinking goes here (analysing the situation, planning)
</inner_monologue>
<run_ipython>
This is where you implement the first step of your plan by running a cell in iPython.
</run_ipython>

-------
Steps to Answer the Question:

1. Read the question carefully.
2. Choose appropriate tools. Read the documentation of the chosen tools to understand the functions of them.
3. Load the graph data.
4. Retrieve the information that is asked in the question by using the provided tools rather than writing code by yourself.
5. Your final answer should be a string only containing the answers to the question without explanatory text.

-------
The answers to the questions have been ensured to be accessible via the provided tools. Do not attempt to use custom programmatic
↪  solutions to extract information from the graph.

-------
At each step, use only one tool, and carefully observe the output of each step to better understand how the tool functions.

-------
Note: Provide only the value of the answer as the final output, without any additional text or full sentences.

-------
Before completing the task, view the final answer to check for potential issues, such as duplicate papers or formatting
↪  inconsistencies.
</guidelines>

<tool_documentation>
[as shown in Appendix A]
</tool_documentation>

<status>
Current date and time: 2025-01-10 20:10:24
Time elapsed since the project beginning: 0:00:00
You are on step 1
Resources spent so far: 0
Number of input tokens remaining: 6897
</status>

Please proceed with your inner monologue to prepare your next action. Strictly follow the XML-like format below:
<inner_monologue>
Thoughts
</inner_monologue>
```

# K    Agent hints

## K.1    Round 1 hints

Our process to generate hints for Round 1 training is the following: first, only a minimal task description and the tool documentation is included in the prompt. Then, the annotator inspects the agent's attempt at a training task. Each time the agent generally understands how to solve the task but makes a minor error, a targeted hint (concise instruction) that directly addresses that particular mistake is added. Each time the agent fails to solve the task entirely, a step-by-step hint is added.

The same process is also used for prompt optimization for our baselines.

**Listing 14** Round 1 hints shared in ToolQA tasks with tabular databases (Airbnb, Coffee, Flight, Yelp).

```
Steps to Answer the Question:

1. Read the question carefully.
2. Load the relevant database.
3. Understand what kinds of information are stored in the database by viewing column names.
4. Retrieve the information that is asked in the question by using the provided tools.
5. Your final answer should be a string only containing the answers to the question without explanatory text.

-------
Directly present the answer in the required format without programmatically converting the output.

-------
Note: Provide only the value of the answer as the final output, without any additional text or full sentences.
```

**Listing 15** Additional round 1 hints for ToolQA-Airbnb tasks.

```
This task is about accommodation data. Make sure to load the correct database.

-------
Hint on solution:
Question: What is the host's name for Amazing MODERN Apartment in Prime Brooklyn in Bushwick?
Hint: Filter the data to get the row where 'NAME' is 'Amazing MODERN Apartment in Prime Brooklyn' and 'neighbourhood' is 'Bushwick'.
Note that the words after the last 'in' in the question indicates the 'neighbourhood', and the words before the last 'in' indicates
↪  the 'NAME'.
Use the exact phrasing from the question to filter the data.

-------
Before completing the task, you must print the value you get to recheck the format of the answer to ensure it is correct.
```

**Listing 16** Additional round 1 hint for ToolQA-Coffee tasks.

```
Before ending the task, print the final answer to check the value and the format of it.
```

**Listing 17** Additional round 1 hints for ToolQA-Flight tasks.

```
The flight identifier is a combination of the 'IATA_Code_Marketing_Airline' (a 2-letter code) and the
↪  'Flight_Number_Marketing_Airline'.
For example:
IATA_Code_Marketing_Airline = 'G4', Flight_Number_Marketing_Airline = '567': 'G4567'
IATA_Code_Marketing_Airline = 'F9', Flight_Number_Marketing_Airline = '124': 'F9124'

-------
Hint on solution:
Question: What was the departure time of the AB123 flight from CDC to EFE on 2020-01-01?
Hint:
1. Filter the data based on the question.
2. Get the value of 'DepTime' for the row (e.g.,'1234.0') into a variable `dep_time`.
3. You must view the value by `print(dep_time)`.
4. Convert the departure time into the 'HH:MM' format directly. For example:
- '840.0' becomes '8:40'.
- '1842.0' becomes '18:42'.

Note: Do not programmatically convert the value. Instead, directly write the answer in the correct 'HH:MM' format.

-------
Use `.format()` to round the answer to the required decimal places if needed.
```

**Listing 18** Additional round 1 hints for ToolQA-Yelp tasks.

```
This task is about business data. Make sure to load the correct database.

-------
In one cell, do not perform too many steps.
```

**Listing 19** Round 1 hints for ToolQA-Agenda tasks.

```
Steps to Answer the Question:

1. Read the question carefully.
2. Use the tool to query the database. Review the tool's documentation to ensure proper usage.
3. When using the tool, write an appropriate query to retrieve relevant documents, and specify the number of documents to retrieve
↪  based on the complexity of the question.
4. Analyze the retrieved documents to find the information needed to answer the question.
5. If the answer is not found, you can refine your query or retrieve more documents.
6. Your final answer should be a string only containing the answers to the question without explanatory text or context.

-------------------
Start by retrieving, e.g., 10 documents.

-------------------
When a large number of documents are retrieved, it may not be feasible to display all of them at once, as the output could be
↪  truncated.

-------------------
Hint: When answering questions about the location of an event, provide the exact name of the place as stated in the document, without
↪  any context. For example, if the document indicates that the event took place at 'the garden of The Central Park', the answer
↪  should be 'the garden of The Central Park' but not 'Central Park'.

-------------------
Dates in the documents are written in words. For example, '04/05' appear as 'April 5'.

-------------------
Note: Provide only the value of the answer as the final output, without any additional text or full sentences.

-------------------
Directly present the answer in the required format without programmatically converting the output.
```

---

**Listing 20** Round 1 hints for ToolQA-DBLP tasks.

---

```
Steps to Answer the Question:

1. Read the question carefully.
2. Choose appropriate tools. Read the documentation of the chosen tools to understand the functions of them.
3. Load the graph data.
4. Retrieve the information that is asked in the question by using the provided tools rather than writing code by yourself.
5. Your final answer should be a string only containing the answers to the question without explanatory text.

-------
The answers to the questions have been ensured to be accessible via the provided tools. Do not attempt to use custom programmatic
↪   solutions to extract information from the graph.

-------
At each step, use only one tool, and carefully observe the output of each step to better understand how the tool functions.

-------
Note: Provide only the value of the answer as the final output, without any additional text or full sentences.

-------
Before completing the task, view the final answer to check for potential issues, such as duplicate papers or formatting
↪   inconsistencies.
```

---

**Listing 21** Round 1 hints for OfficeBench tasks (30 most common hints included; 1 / 2).

---

```
Print the file content and manually read the text to extract the required information. Do not extract it programmatically.

-------
When creating a new file, store it in `data_path`. For example, if the file name is `test.docx`, use `os.path.join(data_path,
↪   "test.docx")` as the file path. Do not use `test.docx` as the file path directly, as this will not save the file in the correct
↪   task files folder.

-------
Do not obtain cell values programmatically for calculations / comparison. Instead, manually inspect the file content to find the
↪   required values before performing calculations / comparison.

-------
Use `print(os.listdir(data_path))` to see the files in the folder.

-------
The OCR tool may return some wrong characters. Please attempt to correct any obvious mistakes you spot, such as 'd' replaced by 'a',
↪   or missing characters.

-------
Use `excel_read_file` to view the file content (you can print it) and manually identify the row and column id to work on. Do not
↪   determine the indices programmatically.

-------
Before completing the task, use `excel_read_file` to view the file content to verify that the file content is correct.

-------
If the data_path directory does not exist, please create it using os.makedirs(data_path).

-------
Perform only one action in one step.

-------
Do not forget import `os`.

-------
Before completing the task, use `word_read_file` to view the file content to verify that the file content is correct.

-------
After creating a new Word file, always remember to write the information into the new Word file using `word_write_to_file()`.
```

---

---

**Listing 21** Round 1 hints for OfficeBench tasks (30 most common hints included; 2 / 2).

```
When reading from data_path, use 'os.path.join(data_path, "file_name")' to set the full path of the file with the correct separators.

-------
After creating a new Excel file, always remember to write the information into the new Excel file using 'excel_set_cell()'.

-------
The folder contains multiple images. As a first step, use os.listdir(data_path) to list all files in the folder and identify the
↪  correct image file to work on.

-------
To solve this task, you should:
        1. Use 'pdf_read_file()' to read the file content and print it.
        2. Manually check the content to find the required information.
        3. Create a new Word file by 'word_create_new_file()'.
        4. Write the required information into the new Word file using 'word_write_to_file()'.
        5. Convert the new Word file to PDF using 'word_convert_to_pdf()'.
        6. Convert this PDF file to image using 'pdf_convert_to_image()'.

-------
To create a PDF file, first create a Word file and then convert it to PDF using word_convert_to_pdf.

-------
If you get the error 'All strings must be XML compatible: Unicode or ASCII, no NULL bytes or control characters', manually extract
↪  the plain text content from the OCR output to exclude problematic characters. Do not attempt to fix the OCR output
↪  programmatically.

-------
You must strictly follow it: In your first step, import 'os' and use 'os.listdir(data_path)' to list all files in the folder. Do not
↪  assume the file name. The file name is not 'Concert Announcement.pdf'

-------
Do not parse the excel file programmatically. Instead, manually check the file content to find the required information.

-------
Verify the events have been created in the calendars before completing the task.

-------
Do not parse the excel file programmatically.

-------
Before read any files, in your first step, import 'os' and use 'os.listdir(data_path)' to list all files in the folder.

-------
Use the list_emails() tool to list the email to get the email id, then use the read_email() tool to display the content of the email.

-------
In the report, you should include the keyword "revenues" as well as the correct years you identified in the content.

-------
To obtain an image, you can first convert the excel file to pdf using 'excel_convert_to_pdf()', and then convert the pdf to image
↪  using 'pdf_convert_to_image()'.

-------
Manually summarize key points of the paper based on its abstract. Print it to check the content before writing it into a word file.

-------
For file operations, import 'os' in your first step.

-------
Information about house price index are stored in an excel file.

-------
The event summary should be detailed and be exactly as mentioned in the task description. For this task, you need to mention the
↪  place of the class in the event summary.
```

## K.2 Round 2 hints

The corrective hints in Rounds 2 and 3 appear only in specific states in which the agent makes each kind of mistake, as localized by the LLM reviewer, not as a contiguous list as shown in the listings below.

---

**Listing 22** Example round 2 corrective hints for ToolQA (1 / 2).

```
After loading the graph by ‘GRAPH_DATA = load_graph()‘ you can use these tools to check nodes / neighbours:
‘check_nodes(GRAPH_DATA, graph_type: str, node: str)‘
    Don't forget any argument: 1. graph data; 2. graph type, which is either either 'PaperNet' or 'AuthorNet'; 3. node name.
‘check_neighbours(GRAPH_DATA, graph_type: str, node: str)‘
    Don't forget any argument: 1. graph data; 2. graph type, which is either either 'PaperNet' or 'AuthorNet'; 3. node name.

-------
When keeping only rows with 'categories' containing a certain business, for example Restaurants business, do not use ‘data_filter()‘.
Instead, always remember to use the following codes: ‘data_business = data[data['categories'].str.contains('Restaurants')]‘.

When filter the data by 'city'/'name', you can use ‘data_filter()‘.

-------
To answer this question, strictly follow these steps:
1. Filter the data only by 'neighbourhood'. Do not do it by any other columns. Then, get values of 'review rate number' of filtered
↪  rows: ‘rates = get_value(filtered_db, 'review rate number')‘
3. Convert to a list of floating-point numbers: ‘rates_list = [float(rate) for rate in rates.split(', ')]‘.
4. Calculate the number of reviews with a rate greater than 4.5: ‘num = sum([1 for i in range(len(rates_list)) if rates_list[i] >=
↪  4])‘.

-------
Hint on solution:
- Calculate the maximum and minimum latitude and longitude within a 5-mile radius from the business mentioned in the question, you
↪  should strictly follow these steps:
- Get the value of 'latitude', 'longitude' and 'review_count' for all businesses into variables ‘lats‘, ‘lons‘ and ‘review_counts‘;
-
‘lats_list = [float(lat) for lat in lats.split(', ')]‘ # convert to float
‘lons_list = [float(lon) for lon in lons.split(', ')]‘ # convert to float
‘review_counts_list = [int(review_count) for review_count in review_counts.split(', ')]‘ # convert to int
- keep only the review counts of businesses within a 5-mile radius from 'Perenn in Bay'. Remember you should not use ‘data_filter()‘
↪  for this step.
Strictly follow this: ‘review_counts_within_5 = [review_counts_list[i] for i in range(len(review_counts_list)) if
↪  (lats_list[i]<=max_lat and lats_list[i] >= min_lat and lons_list[i] <= max_lon and lons_list[i] >= min_lon)]‘
- Calculate the average review counts. Make sure the final answer is in a correct format.

-------
You should get the value of 'ArrDelayMinutes'.

-------
To determine available time windows from 9:00 AM to 6:00 PM for a meeting:
1. Read the retrieved document carefully to identify event(s) between 9:00 AM and 6:00 PM. Focus on identifying any time slots where
↪  events conflict with the potential meeting time.
2. Use reasoning in your monologue to determine available time windows. The answer should be all time slots within 9:00 AM to 6:00 PM
↪  where no event is scheduled.
    Note: Do not use programmatic solutions to calculate or extract the answer. Consider only available time windows within 9:00 AM
    ↪  to 6:00 PM.
    Example:
    If an event is from 9:30 AM to 1:00 PM, the answer would be: 9:00 AM-9:30 AM, 1:00 PM-6:00 PM.
    If an event is from 4:00 PM to 6:00 PM, the answer would be: 9:00 AM-4:00 PM.
    If an event is from 7:00 AM to 10:00 AM, the answer would be: 10:00 AM-6:00 PM.
    If an event is from 6:00 PM to 7:00 PM, the answer would be: 9:00 AM-6:00 PM. (Events beyond the range of 9:00 AM to 6:00 PM
    ↪  should not affect the answer.)
3. Check your answer. If you are sure that your answer is correct in both value and format, you can call ‘complete_task(final_report:
↪  str, return_value)‘ to finish the task.
```

---

---

**Listing 22** Example round 2 corrective hints for ToolQA (2 / 2).

```
Check if "ByAppointmentOnly" is in attributes: If it does and 'ByAppointmentOnly' is 'True', the answer should be 'Yes'; If it does
↪  not have 'ByAppointmentOnly', or 'ByAppointmentOnly' is 'False', the answer should be 'No'.
Do not make your judgement based on other attributes.

-------
This question is related to accommodation. You should load the airbnb database.

-------
To answer this question, strictly follow these steps:
1. Filter the data only by 'FlightData'. Do not do it by any other columns.
2. Get the value of 'Distance' for filtered rows: `distances = get_value(filtered_db, 'Distance')`
3. Convert to a list of floating-point numbers: `distances_list = [float(dist) for dist in distances.split(', ')]`.
4. Calculate the number of flights with a distance greater than 300 miles: `num = sum([1 for i in range(len(distances_list)) if
↪  distances_list[i] > 300.0])`.

Solve this question in this cell.

-------
After you get the time, do not programmatically convert the value. Instead, directly write the answer in the correct format.
For example:
- If you get '820.0', then the answer is '8:20'. (Note: Do not write as '08:20')
- If you get '1042.0', then the answer is '10:42'.
```

---

**Listing 23** Round 2 corrective hints for OfficeBench (1 / 2).

```
When the task requires you to extract information from a file for purposes such as comparison, calculation, analysis, or writing to
↪  another file, you must manually inspect the file content and do reasoning like a human. Do not programmatically parse the file
↪  content (e.g., with pattern-matching). Also, when writing to a file, you should use the file-writing tool. Do not use loops to
↪  write the content.

-------
Use `send_email()` correctly without extra arguments. `send_email()` strictly accepts four string arguments: sender's email address
↪  (e.g., alice@example.com; the sender typically is the user you are working for), recipient's email address (e.g.,
↪  bob@example.com), subject (used to generate the email file name), and content.
Do not: 1. add one extra argument for the attachment. If the task involves sending an attachment, mention the attachment in the email
↪  content. 2. Use invalid email formats. Email addresses must follow the standard format, like user@example.com. Do not use .eml
↪  at the end of the address.
The generated email file will be named {subject}.eml. So if the required file name is e.g., bob.eml, the subject must be exactly
↪  'bob' (without .eml).

-------
When you read the content of a word/excel/pdf file using file-reading functions or extract text content of an image using OCR, and
↪  store the content to a python variable, e.g. `content = pdf_read_file(file_path)`, you must remember to print the variable
↪  `content`. This allows the content to be inspected, referenced, or used in further reasoning steps.

-------
Always read the task description carefully and follow every requirement precisely.
Pay special attention to the following:
- File creation: If the task asks you to create a new file (e.g., a report or .txt file), you must create it. Do not skip this step.
- File type: If the task task specifies a file type, like a txt file or an image file, you must create that exact typedo not use a
↪  different format.
- File name: Use the exact file name provided in the task. Do not invent or modify the name.
- File content: If specific content must be written into a file, write it exactly as provideddo not change, omit, or add anything.

-------
Use `list_email(username)` to get all emails' ids. Then, you must use `read_email(username, email_id)` to read all emails one by one.
Do not assume the target email is the first one in the listemails are not in chronological or predictable order.
Do not use loops (e.g., for or while) to iterate over emails. Instead, explicitly call read_email() for each email id one at a time.
When inspecting email content, do not parse it programmatically. Instead, read the content as a human would to extract the needed
↪  information.
```

---

**Listing 23** Round 2 corrective hints for OfficeBench (2 / 2).

```
You are only allowed to use these built-in Excel functions for working on excel files:
: `excel_create_new_file()`, `excel_delete_cell()`, `excel_set_cell()`, `excel_convert_to_pdf()`, and `excel_read_file()`.
Do not use undefined or imaginary functions, such as: `excel_add_column()`, `excel_delete_file()`, `excel_read_cell()`.
Note: `excel_read_file()` returns the entire content of an excel file. **It cannot be used to access individual cells/rows/columns.**
Also, do not programmatically parse the content of the excel file. Instead, read the content as a human would to extract the needed
↪  information.

-------
Use the function `create_event()` to create a calendar event as a reminder.
You must provide four arguments in the exact order: the user name, the event summary, the start time, and the end time.
The event summary must be detailed, including these information if they are available: the introduction/name of the event, who is the
↪  event for/who participate in the event/who hold the event, where the event is held. The words in the summary should match the
↪  original wording in the task description and task file.
For the start and end times, use the exact time given in the task or the task file. Do not use datetime.now() or generate your own
↪  time.

-------
When the task requires generating a pdf file (e.g. summarize information in a PDF file), follow these process: create a new word file
↪  using `word_create_new_file()`; write the content to the word file using `word_write_to_file()`; convert the word file to a pdf
↪  using `word_convert_to_pdf()`.
Do not use undefined or imaginary functions, such as: `excel_convert_to_word()`, `pdf_write_to_file()`.

-------
When the task file is an image and you need to read text of the image, use `ocr_recognize_text()`. Do not convert the image to a pdf
↪  first, this step is unnecessary and incorrect.
While to read text of a pdf file, use `pdf_read_file()`. Do not use `ocr_recognize_text()` to read a pdf content, this function is
↪  only for image files.

-------
When the task requires generating an image file, follow these process: create a new word file using `word_create_new_file()`; write
↪  the content to the word file using `word_write_to_file()`; convert the word file to a pdf using `word_convert_to_pdf()`; convert
↪  the pdf file to an image using `pdf_convert_to_image()`.
Do not use undefined or imaginary functions, such as: `excel_convert_to_image()`, `word_convert_to_image()`, `pdf_write_to_file()`.

-------
Note that `data_path` is a read-only variable, avoid modifying it. Use `print(os.listdir(data_path))` to see the files in the folder.

-------
This task requires you to answer a question and provide a final answer. In this step, when using `complete_task(final_report,
↪  return_value)`, make sure to set return_value to the final answer. The return_value must be a string, e.g. complete_task(""The
↪  age is:", "10").
```

### K.3 Round 3 hints

---

**Listing 24** Example round 3 corrective hints for ToolQA (1 / 2).

```
To get the number of reviews, perform the following operations in the next step:
1. Load the airbnb database;
2. Then, filter the data to by 'NAME' 'neighbourhood' and 'id'.
3. Get the value of 'number of reviews'. Print it to view the value. Return it in the correct format.

Think in your monologue about above operations.
Perform all above operations in the next cell.

-------
First, you should load the graph data: 'GRAPH_DATA = load_graph()'.

-------
Perform the following operations in the next step:
1. Filter the data to get the row where 'room type' is 'Shared room'. Always remember to filter by "room type" instead of "room_type".
Then get values of 'latitude', 'longitude', 'price' of filtered rows into variables 'lats', 'lons', 'prices'.
2. '''# strcitly follow this
lats_list = [float(lat) for lat in lats.split(', ')]
lons_list = [float(lon) for lon in lons.split(', ')]
prices = prices.replace('$','').replace(',','')
prices_list = [float(price) for price in prices.split()]'''
3. Find the index of the minimum price, also within a 10 miles radius from the given point. Strictly follow this:
'''
min_index = -1
for i in range(len(prices_list)):
    if (lats_list[i]<=max_lat and lats_list[i] >= min_lat and lons_list[i] <= max_lon and lons_list[i] >= min_lon):
        if min_index == -1 or prices_list[i] < prices_list[min_index]:
            min_index = i
'''
4. Use the index to get the name. Do it strcitly as follows: 'list(filtered_db['NAME'])[min_index]'. Remember the column name is
↪  'NAME'.

-------
Now that you got a list of co-authors, you should perform the following operations in the next step:
'''
max_citations = 0
answer = []
for co_author in co_authors:
    papers = [paper for paper in check_edges(GRAPH_DATA, "AuthorNet", author, co_author)["papers"]] # find papers co-authored by the
    ↪  author and co-author
    papers = list(set(papers)) # always remember to remove duplicates
    for paper in papers:
        citation += check_nodes(GRAPH_DATA, "PaperNet", paper)["number of citations for this paper"]
    if citation > max_citation:
        max_citation = citation
        answer = [co_author]
    elif citation == max_citation: # multiple collaborators with the same citations
        answer.append(co_author)
answer = "; ".join(answer) # If there are multiple collaborators with the same citations, separate them with semicolon and space '; '
'''

-------
In the next step, you should:
1. Convert airtime you get to a list of floating-point numbers: 'air_times_list = [float(time) for time in air_times.split(', ')]'.
2. Remove the 'nan' values from the 'air_times_list'. Do it strictly as follows: 'air_times_list_nonnan = [time for time in
↪  air_times_list if time >= 0]'. Keep only non-negative values to ensure that there is no 'nan' value.
Perform above operations in the next step.

-------
You should print the value, then directly write it in the correct format. Do not convert it programmatically.
```

---

---

**Listing 24** Example round 3 corrective hints for ToolQA (2 / 2).

```
The flight identifier is a combination of the 'IATA_Code_Marketing_Airline' (a 2-letter code) and the
↪ 'Flight_Number_Marketing_Airline'.
For example:
IATA_Code_Marketing_Airline = 'G4', Flight_Number_Marketing_Airline = '567': 'G4567'
IATA_Code_Marketing_Airline = 'DL', Flight_Number_Marketing_Airline ='1271': 'DL1271'
'IATA_Code_Marketing_Airline' has 2 letters. Digits after 2-letter code is 'Flight_Number_Marketing_Airline'.
Therefore, to find the average flight time of G4886, we need to first filter the database to get rows where
↪ 'IATA_Code_Marketing_Airline' is 'G4' and 'Flight_Number_Marketing_Airline' is '886':
'flights_db = load_db('flights')'
'filtered_db = data_filter(flights_db, 'IATA_Code_Marketing_Airline=G4; Flight_Number_Marketing_Airline=886')'

-------
After you get the value of 'TaxiIn', you should print it to view the value. Then, directly write it in the correct format. Do not
↪ convert it programmatically.

-------
The attribute "number of citations for this paper" means the number of papers that cited this paper, so it is the answer to the
↪ question.

-------
To answer this question, you must follow the example solution:
1. Retrieve 1000 documents related to this person;
2. Always remeber to filter again to make sure these documents are related to this person: 'docs_with_person = [doc for doc in
↪ retrieved_docs if 'This Person' in doc]';
3.Always remember to filter for documents that mention the specific date in the question. Dates must strictly follow the format:
↪ "Month in word + day in number" (e.g., "January 1").
For example: docs = [doc for doc in docs_with_person if 'January 1' in doc]
4. Print the filtered documents to view them.
# Ensure that you have followed the example solution correctly. Continue adhering to these steps in your next actions.

-------
Perform the following operations in the next step:
1. Get the value of 'Distance' and 'AirTime' for filtered rows into seperate variables: 'distances', 'air_times'.
2. Convert to lists of floating-point numbers:
'distances_list = [float(dist) for dist in distances.split(', ')]'
'air_times_list = [float(time) for time in air_times.split(', ')]'
3. Always remember to remove both the 'nan' values from the 'air_times_list', and corresponding values in 'distances_list'.
'air_times_list_nonnan = [air_times_list[i] for i in range(len(air_times_list)) if air_times_list[i] >= 0]'
'distances_list_nonnan = [distances_list[i] for i in range(len(air_times_list)) if air_times_list[i] >= 0]' # Do not forget to remove
↪ corresponding values from 'distances_list' as well.
5. Calculate the average speed: 'avg_speed =sum(distances_list_nonnan) / sum(air_times_list_nonnan)'.

-------
After you get the value of "ArrDelayMinutes", you should print it to view the value.

-------
Since you retrieved only 10 documents, you should print them to view the content.

-------
If the event is outside the 9:00 AM to 6:00 PM. This means that the entire window of 9:00 AM to 6:00 PM is available for the meeting.
↪ Thus, the answer is "9:00 AM-6:00 PM".

-------
Based on the question, to get the number of reviews, you should filter the data by "NAME", "neighbourhood" and "id". Do not forget
↪ filtering by "id".
```

---

---

**Listing 25** Round 3 corrective hints for OfficeBench.

```
When sending an email, if detailed information is available (from the task or the file), include it in the email body. If the task
↪   requires sending a file, do not simply say The file is attached. Instead, include the full content of the attched file in the
↪   email, as long as the information is available.

-------
When the task requires you to extract information from a file for purposes such as comparison, calculation, analysis, or writing to
↪   another file, you must manually inspect the file content and do reasoning like a human. Do not programmatically parse the file
↪   content (e.g., with pattern-matching).

-------
When the task file is a pdf file, use `pdf_read_file()` to read the file.

-------
Always read the task description carefully and follow every requirement precisely.
Pay special attention to the following:
- File creation: If the task asks you to create a new file (e.g., a report or .txt file), you must create it. Do not skip this step.
↪   Even if you don't know what to write in, you should still create the file and says, e.g., "no content needed" in the file.
- File name: Use the exact file name provided in the task. Do not invent or modify the name.
- File content: If specific content from the task description or the task file must be written into a file, write it exactly as
↪   provided in the task or the task filedo not change, omit, or add anything.

-------
When setting calendar event times, if a specific time is provided in the task files, you must use that exact time. If the time is not
↪   in the files, the calendar time should be set based on the that stated date. For example, if the task description says "Today is
↪   2020-05-01", then the year of the calendar time should be set to 2020 (do not assume the current year). Never use datetime.now()
↪   or generate/assume your own time.

-------
When you read the content of a word/excel/pdf file using file-reading functions or obtain users' emails, and store the content to a
↪   python variable, e.g. `content = pdf_read_file(file_path)`, you must remember to print the variable `content`. This allows the
↪   content to be inspected, referenced, or used in further reasoning steps.

-------
Note that `data_path` is a read-only variable, avoid modifying it. Use `print(os.listdir(data_path))` to see the files in the folder.

-------
Use `list_email(username)` to get all emails' ids. Then, you must use `read_email(username, email_id)` to read **all emails one by
↪   one**.
Do not assume the target email is the first one in the listemails are not in chronological or predictable order.
Do not use loops (e.g., for or while) to iterate over emails. Instead, explicitly call read_email() for each email id one at a time.

-------
`excel_set_cell()` typically only takes four arguments: the file path, the text, the row index and the column index. Do not need to
↪   specify the sheet name.

-------
When the task file is an excel file, use `excel_read_file()` to read the file.

-------
When the task file is an image, you need to read text of the image using `ocr_recognize_text()`. Do not convert the image to a pdf
↪   first, this step is unnecessary and incorrect.

-------
`word_write_to_file()` takes only two arguments: the file path and the content. Do not add any other arguments, such as `append`.
```

## L   Example prompt for the LLM-based reviewer

The LLM reviewer was implemented with the same model as the teacher (the trained Llama-3.1-70B based agent from the previous round) for memory efficiency, i.e., to avoid serving multiple 70B models at training data generation time.

**Listing 26** An example prompt for the LLM-based reviewer for detecting the mistake of *not directly returning the number of citations* in ToolQA agent trajectories.

```
You are provided with the trajectory from an AI agent. The agent's task is to answer a question by retrieve information from Graph
↪   data. The agent performs this task by thinking in inner monologues and writing Python code to execute its actions.
The agent needs to determine how many papers have cited a specific paper mentioned in the question. The correct way to solve this is
↪   to check the "number of citations" attribute for the paper in question.

Your task is to determine whether the agent has followed the correct solution or if it has made a mistake in the last step.
↪   Specifically, the agent may sometimes overcomplicate the solution and fail to directly return the value of "number of citations"
↪   as the final answer.

Here is the format of the response you should strictly follow:
    ----
    <reasoning>
    Write down your reasoning here.
    </reasoning>
    <answer>
    Write down your answer here.
    Your answer should be either True or False.
    </answer>
    ----

Answer True if the agent has strictly followed the correct solution (i.e., directly returning the "number of citations for this
↪   paper"); or the last step is not related to the task (for example, graph loading or calling `complete_task()` to end the task).
Answer False if the agent made a mistake in the last step (i.e., overcomplicating the solution or failing to return the citation
↪   count directly).

Below is the agent's trajectory. Please make your judgment based on the last step only. The earlier steps are included as additional
↪   context.
```

