# OpenReview forum: "Memento No More: Coaching AI Agents to Master Multiple Tasks via Hints Internalization"
_TMLR — Decision pending for TMLR_

### Review · Reviewer_Jn6j · 2025-12-18

**Summary Of Contributions:**

This paper proposes an AI agent training method that can handle multiple tasks through hint internalization. Existing approaches rely on prompts to provide task-specific knowledge, but they do not generalize well to multiple tasks. In contrast, the proposed method internalizes knowledge across multiple tasks via a human-in-the-loop iterative process, leveraging human-provided hints to learn multi-task information. Experimental results demonstrate the effectiveness of the proposed method.

Strengths:
1. The paper is well organized and easy to follow.
2. The proposed method shows consistent improvements over baseline approaches, demonstrating its effectiveness.

Weaknesses:
1. A major concern is that the training pipeline requires human involvement, which makes the method less practical for real-world applications. An automatic pipeline would be much more appreciated.
2. The experimental comparison does not include existing related approaches. Only self-designed task-specific-hint and combined-hint baselines are considered. Comparing with existing approaches (even if the settings are not identical, similar methods could be adapted) would strengthen the claims.
3. The experiments are conducted using only one agent/LLM (LLaMA-3.1-70B), which is insufficient. Evaluations on additional LLMs are needed to demonstrate the generalizability of the proposed method.
4. Figure 3 is difficult to understand.

**Audience:**

Yes

**Audience Explanation:**

Yes, audiences from the AI agent domain who care about multi-task generalization would be interested in this work.

**Broader Impact Concerns:**

The broader impact statement is well-written. I have no concern.

**Claims And Evidence:**

Yes

**Claims Explanation:**

Yes, the experimental results clearly demonstrate the effectiveness of the proposed method.

However, comparing with existing related approaches and applying the method to additional LLMs would make the claims more convincing.

**Requested Changes:**

1. Compare with existing related approaches
2. Apply the method to additional LLMs
3. Make Figure 3 easier to understand
4. Propose an automatic pipeline without human involvement, or at least discuss a potential solution

---

> ### Author Response · Authors · 2026-02-22
> **Response to Reviewer Jn6j**
>
> We thank the reviewer for the thorough and detailed feedback. Below we address each of the requested changes and summarize the revisions incorporated into the updated manuscript.
>
> ### **Comparison to related methods**
>
> In addition to the task-specific teacher and combined-hint baselines, we include a comparison to SFT on the hint-powered trajectories, i.e., standard cross-entropy loss on the teacher’s sampled tokens (Section 5.4; Table 7 in Appendix B.2; also included below). This kind of SFT loss is analogous to the approach taken in prior trajectory fine-tuning work AgentBank (Song et al., 2024). Our proposed KL distillation outperforms this approach both on average and individually in 5/6 of the ToolQA domains:
>
> | Loss | Trajectories | Round | DBLP | Agenda | Yelp | Airbnb | Flight | Coffee | Avg. |
> |-|-|-|-|-|-|-|-|-|-|
> | KL | all | 1 | 92.1 | 94.4 | 84.7 | 89.5 | 92.2 | 97.7 | 91.8 |
> | KL | all | 2 | 97.7 | 92.9 | 94.9 | **96.1** | 93.3 | **99.5** | 95.7 |
> | KL | all | 3 | **98.9** | 96.0 | **98.9** | **96.1** | **98.3** | **99.5** | **97.9** |
> | cross-entropy (hint-powered SFT) | success only | 1 | 94.9 | **99.2** | 84.2 | 88.2 | 91.1 | 95.8 | 92.2 |
> | cross-entropy (hint-powered SFT) | success only | 2 | 97.7 | 96.8 | 88.7 | 86.9 | 93.3 | 98.6 | 93.7 |
> | cross-entropy (hint-powered SFT) | success only | 3 | 97.7 | 97.6 | 97.7 | 93.5 | 96.1 | 97.7 | 96.7 |
>
> To make this baseline comparison clearer, we have updated Section 5.4 to explicitly highlight the connection to SFT and AgentBank. The key distinction from AgentBank is that it relies on trajectories collected from a stronger teacher model (GPT-4), whereas our method distills from the same model augmented with targeted human hints, making it applicable in settings where no stronger model is available.
>
> Beyond this, we are not aware of multi-task agent training methods that share sufficient setup similarity to make an apples-to-apples comparison. We would welcome specific suggestions from the reviewer if they have particular methods in mind. Many prior works, such as Reflexion (Shinn et al., 2024) and ExpeL (Zhao et al., 2024) operate without any weight updates, and instead store experience in prompting-based memory buffers. Rather than a substitute for such inference-time designs, our training method is complementary: by correcting the base model's behavior through weight updates, it eliminates the need to re-address the same mistakes at inference time, and could equally be combined with memory- or RAG-style approaches where dynamic information retrieval is needed.
>
> ### **Figure 3**
>
> Thank you for pointing out the room for improvement in Figure 3, we have redesigned the figure in the updated manuscript to improve its clarity. Specifically, we simplified the trajectory diagram in the left panel (Round 1). We have also applied the magenta highlight color consistently to all actions informed by the teacher's additional context, including in the conversation history in Round 1. We hope these changes improve the readability of the figure.
>
> ### **Experiments with another model**
>
> We agree that experiments with an additional LLM would strengthen the generalizability claim, and we appreciate the reviewer raising this. Unfortunately, training a full multi-round agent on an additional 70B-scale model was not feasible within the revision timeline. That said, we note that the core components of our method (context distillation via KL divergence, LoRA adaptation, and LLM-based mistake localization) are all model-agnostic in design, and we expect the training pipeline to transfer to other instruction-tuned LLMs without fundamental modification. We plan to include experiments on at least one additional model in future work.
>
> ### **Feasibility of an automatic annotation pipeline**
>
> We appreciate this feedback. While our primary focus is on designing a practically applicable, annotation-efficient framework for human supervision, a fully automated pipeline is a natural and valuable direction for future work.
>
> Some concrete possibilities include:
>
> - LLM-as-annotator, where a larger / more capable LLM replaces the human review step by identifying failure patterns and writing corrective hints automatically.
> - Environment-driven self-critique: In settings with rich environmental feedback (e.g., code execution errors, failed API calls, automated task verification), an agent could self-critique without external oversight.
>
> However, both approaches rely on preconditions that do not always hold: that a more capable agent already exists, or that the environment can provide informative failure signals. Our method is designed to be applicable in the more general case where neither condition is guaranteed. Thank you for rasining this point -- we have expanded the discussion on possible automization of the pipeline in the limitations section of the revised manuscript accordingly.

---

### Review · Reviewer_XqXH · 2025-12-31

**Summary Of Contributions:**

This paper studies a practical failure mode in multi-task LLM agents: when practitioners aggregate “all possible” task hints into a single long prompt (because it’s hard to know which hint will matter at a given step), agents must re-read a convoluted prompt at every step, which can degrade performance and increase cost. The authors motivate this as an “anterograde amnesia / notes” analogy and explicitly point out prompt-length overhead (including quadratic attention cost).

To address this, they propose **Memento No More (MNM)**: an iterative **human coaching + hint internalization** procedure that converts prompt-based guidance into model weights via **context distillation** (teacher has hints; student is trained to behave as if it had them, without seeing them). Concretely:

- Agent setup: a ReAct-style agent produces (i) an “inner monologue” and (ii) executable Python/tool calls; tool docs and best practices initially live in the prompt.
- Prompt-engineer initial hints (tools + generic best practices), collect teacher trajectories with hints, then distill them into a LoRA-updated student so it no longer needs task-specific hints in the prompt.
- Analyze the trained agent’s failures, detect mistake states (via scripts and/or LLM-based review), write targeted corrective hints for those states, and distill again—iterating until mistakes are addressed.

Empirically, they show a “Memento effect” where combining all hints into one prompt causes notable success-rate drops vs task-specific prompting, and they report that MNM training recovers and improves performance, reaching strong success rates without hints and competing with or surpassing larger/proprietary models on their benchmarks.

**Audience:**

Yes

**Audience Explanation:**

The paper sits at the intersection of LLM agents, human feedback, and distillation/continual improvement. It offers a concrete alternative to ever-growing prompts: iteratively distilling human-authored guidance into weights for interactive tool-using agents, with strong benchmark evidence and efficiency implications. This is directly relevant to TMLR readers working on agentic systems, data-efficient alignment/finetuning, and practical deployment constraints (latency/cost from long contexts).

**Broader Impact Concerns:**

None.

**Claims And Evidence:**

Yes

**Claims Explanation:**

* The “Memento effect” is supported by explicit baseline drops when combining hints into a single multi-task prompt on both benchmarks.
* The efficacy of MNM is backed by round-by-round success-rate tables for ToolQA and OfficeBench showing improvements up to round 3, including comparisons against DeepSeek-V3 and GPT-4o in the same agent setting.
* Generalization is partially addressed via held-out ToolQA templates (Table 5), where MNM improves substantially over round 1 and remains competitive with strong baselines.
* Efficiency claims are supported by a token-usage table showing an order-of-magnitude reduction in input tokens for MNM round 3 relative to prompt-heavy baselines.
* The paper also includes ablations (dropout; data balancing) that support key design choices.

**Requested Changes:**

Overall, the authors have done a commendable job, and the following minor suggestions may further help readers deepen their understanding:
* Stronger analysis of the reviewer/LLM-judge pipeline: estimate false positive/negative rates for mistake localization, and sensitivity of results to reviewer quality (script vs LLM vs hybrid).
* Broaden evaluation / transfer: test whether hints distilled on one benchmark/domain help or hurt another, or add at least one more domain to assess generality.
* Clarify training–deployment tradeoffs: summarize training cost vs inference savings in the main text (token reductions are compelling, but readers will want a clearer “break-even” story).

---

> ### Author Response · Authors · 2026-02-22
> **Response to Reviewer XqXH**
>
> We thank the reviewer for the careful reading and generous assessment. Below we address each of the three requested changes and summarize the corresponding revisions in the updated manuscript.
>
> ## **Stronger analysis of the reviewer/LLM-judge pipeline**
> We have added a new analysis (included in the updated Appendix E and below) characterizing the accuracy of our automated mistake-localization pipeline across all hint types in both rounds of training. For each filter, we report the false positive rate (FPR; fraction of correct trajectories incorrectly flagged as containing a mistake) and the false negative rate (FNR; fraction of mistake-containing trajectories missed by the filter):
>
> **OfficeBench:**
>
> Average over LLM-based filters:
> - Round 2: FPR=0.6% FNR=0.3%
> - Round 3: FPR=0.4% FNR=0%
>
> **ToolQA:**
>
> There are 4 types of mistakes and 2 types of mistakes can be identified using script-based filters in round 2 and round 3, respectively.
> Average over LLM-based filters:
> - Round 2: FPR=0.3% FNR=0.3%
> - Round 3: FPR=0.1% FNR=0.7%
>
> The breakdown of scores over individual ToolQA hints is included in the new Appendix E.
>
> These low error rates indicate that LLM-based reviewers reliably localize mistakes, with minimal contamination of the training data from either spurious or missed labels.
>
> We also compare LLM-based filters to simpler script-based filters. Script-based detection is possible for 6 mistake types in ToolQA where mistakes can be detected deterministically via string matching or abstract syntax tree checks (full list in Appendix E). For these mistakes, script-based filters achieve a perfect 0% FPR & FNR. However, scripts are inherently more limited: the majority of mistake types involve nuanced reasoning errors (e.g., incorrect post-processing logic, misinterpretation of column semantics, or inappropriate programmatic solutions) that cannot be reliably captured through pattern matching. *No script-based filters are applicable for OfficeBench*, where all mistake types require semantic understanding of task context. Overall, these results suggest that LLM-based filters, which we use throughout our work, provide a practical and accurate solution for mistake localization across the diversity of errors exhibited by the agent.
>
> ## **Transfer of hints to unseen domains**
> We thank the reviewer for this suggestion. To assess cross-domain generalization, we conducted a new held-out experiment in which MNM agents are trained on two ToolQA task groups and evaluated zero-shot on another two. We focus on the four tabular database task groups (Yelp, Airbnb, Flight, Coffee), which share the same underlying tool set, making this a natural testbed for transfer. Results are reported in Appendix B.4 and below:
> |Training domains|Yelp|Airbnb|Flight|Coffee|
> |-|-|-|-|-|
> |Yelp+Airbnb|-|-|45.5|87.3|
> |Airbnb+Flight|63.8|-|-|88.3|
> |Flight+Coffee|64.4|69.9|-|-|
> |Yelp+Flight|-|82.4|-|87.3|
> |Yelp+Coffee|-|77.8|42.2|-|
> |Airbnb+Coffee|63.3|-|50.6|-|
> |**Average**|63.8|**76.7**|46.1|**87.6**|
> |**No ToolQA training**; Single-task agent + task-specific tool hints (Table 1)|**64.4**|50.3|**49.4**|85.4|
>
> This shows that hints distilled for one pair of domains transfer positively to held-out domains in the case of Airbnb and Coffee, with a significant +26.4% to Airbnb. This suggests that the internalized skills, such as correct database loading, filtering, and value extraction, are genuinely transferable across tabular tasks.
>
> Transfer to Flight is more challenging, likely reflecting that flight-specific reasoning (such as parsing flight identifiers and handling delay calculations) requires targeted experience that is harder to acquire from other tabular domains. Overall, these results support the generalizability of our approach within a shared tool ecosystem.
>
> ## **Training-deployment tradeoffs**
> The amortized training cost is indeed relevant to consider when evaluating inference-time efficiency gains. In addition to the token reductions reported in Appendix C, we report a 3-4× inference speedup in Section 5.3. We observed an average runtime of approximately 90 seconds per ToolQA task for the MNM Round 3 agent, compared to 340 seconds for the untrained Llama-3.1-70B with combined hints, when running a vLLM server on 4 AMD MI250X GPUs.  The equivalent numbers for OfficeBench are 140s for our Round 3 agent vs. 600s for Llama-3.1-70B.
>
> The exact durations depend on hardware, but using these numbers as a reference: assuming savings of 250s per task and a total training time of 19.75h (16+1.25+2.5h, Appendix D.3) for ToolQA on identical hardware, the break-even point is 285 task executions, a threshold readily exceeded in any realistic deployment scenario. For OfficeBench, assuming 460s saved per task, the break-even point is even earlier: 32.5h (27+3.5+2h) of training is compensated for by the inference savings for 255 task executions.
>
> We have added the runtime figures and break-even estimate to Appendix C, with a brief summary in Section 5.3.

---

### Review · Reviewer_k2RM · 2026-01-31

**Summary Of Contributions:**

Current LLMs require comprehensive prompts to perform specialized tasks. As the prompt size increases the compute cost increases and performance may degrade.

The main contribution of the paper is to internalize hints provided by a human expert for a specific step of the task on a recurring basis. The paper reports that after a few iterations such a system will outperform the current method of using prompt based hints.

During the initial training state, prompt tuning is used to generate a set of actions for a variety of tasks. The tasks and outputs, is then used to train a student LLM, which has the initial prompt internalized through LoRA adapter.

In subsequent rounds, an expert analyzes the improved model's results to come up with tasks and action specific hints. The teacher agent is updated with this additional hints and the student is trained on the new output from this updated teacher agent.

Strengths:
Inference Efficiency: The method reduces input token counts to 7–10% of standard prompting approaches and improves inference speed by 3–4x.

High Performance: After three rounds of feedback, the Llama-3-based agent outperforms larger proprietary models like GPT-4o and DeepSeek-V3 on complex benchmarks.

Retention of General Skills: Evaluations on HumanEval and GSM8K indicate no significant degradation in the backbone model's general reasoning or coding capabilities.

Key Weaknesses:

Human-in-the-Loop Dependency: The training protocol relies on human expertise to design filters and hints, which may limit scalability for very large task sets.

Domain Specificity: The current experiments are limited to specific tool-use domains (ToolQA and OfficeBench) and do not yet demonstrate cross-domain generalization.

**Audience:**

Yes

**Audience Explanation:**

Yes. The TMLR audience is likely interested in this work for several reasons:

Efficiency: Reducing the scaling costs of LLMs by internalizing long prompts is a significant practical advancement.

Agentic Workflows: The focus on multi-step reasoning, tool use, and sequencing addresses core challenges in the development of autonomous agents.

Human-AI Interaction: The "coaching" metaphor provides a structured way to integrate human feedback into the fine-tuning process beyond simple SFT or RLHF

**Broader Impact Concerns:**

The paper acknowledges potential concerns regarding the automation of knowledge work and the risk of misuse, such as cyberattacks. However, the paper would benefit from a more detailed discussion on the transparency of internalized knowledge. Unlike prompts, which are easily auditable, internalized weights are black boxes, making it harder for users to know exactly which hints or constraints the agent is following during a live task. Clear protocols for logging agent actions could be a mitigation strategy.

**Claims And Evidence:**

Yes

**Claims Explanation:**

The claims are supported by empirical results across multiple benchmarks. The paper reports a 97.9% success rate on ToolQA and a 90.3% rate on OfficeBench, both surpassing GPT-4o. The paper also provides evidence for efficiency with a drastic reduction in input tokens since the model no longer needs tool descriptions in every prompt. The ablation study comparing KL divergence distillation with standard cross-entropy loss justifies the choice of training methodology, showing that distillation leads to better results.

**Requested Changes:**

Quantify Human Effort: Provide a more detailed breakdown of the time required for a human expert to analyze mistakes and design the 18–32 hints used in the experiments. How does it impact the overall cost and scalability of the proposed method.

Comparison with SFT: Include a baseline comparison showing the performance of a model trained via standard Supervised Fine-Tuning (SFT) on the same "hint-powered" trajectories to isolate the benefits of context distillation. This is strongly recommended for recommendation to publish.

Hint Sensitivity: Discuss how sensitive the model is to the quality or phrasing of the human-provided hints. This is strongly recommended for recommendation to publish.

---

> ### Author Response · Authors · 2026-02-22
> **Response to Reviewer k2RM**
>
> We thank the reviewer for the constructive feedback and positive assessment. Below we address the requested points and summarize the revisions incorporated into the updated manuscript.
>
> ### **Quantification of human effort**
> We agree that providing a clearer quantification of human effort is important for evaluating scalability. Below we report the approximate annotation time required per hint based on our experimental workflow.
>
> | Annotation task | Time (min) |
> |-|-|
> | 1. Mistake identification (reading a failed trajectory) | 5 |
> | 2. Reviewer prompt writing | 2 |
> | 3. Reviewer prompt verification (inspecting a subset of filtered steps) | 3 |
> | 4. Hint writing | 2 |
> | 5. Hint verification (inspecting a subset of corrected steps) | 3 |
> | 6. Reviewer prompt refinement | 0.5 |
> | 7. Refined reviewer prompt verification | 1.5 |
> | 8. Hint refinement | 0.5 |
> | 9. Refined hint verification | 1.5 |
> | **Total per finalized hint** | **19** |
>
> This calculation includes any required iterations of reviewer prompt refinement and hint refinement. On average, 0.5 prompt refinement and 0.5 hint refinement iterations were needed for each identified mistake type before the outputs were deemed of sufficient quality (in both the accuracy of filtering and the automatically-evaluated mistake correction rate).
>
> The estimated annotation time is therefore 7.6 and 10.1 hours for 24 and 32 hints, respectively.
>
> Importantly, human effort scales with the number of **distinct mistake types**, not with the number of trajectories or task instances. Once a mistake pattern is identified, automated reviewer filters localize similar failures across trajectories, allowing a single hint to correct many states simultaneously. In an efficient annotation pipeline, this step can happen asynchronously from human annotation time.
>
> We have added this quantitative breakdown to the revised manuscript in Appendix D.4 and refer to this new section in the discussion on human effort in the limitations section to clarify annotation requirements and scalability considerations.
>
>
> ### **Comparison with SFT**
>
> > Include a baseline comparison showing the performance of a model trained via standard Supervised Fine-Tuning (SFT) on the same "hint-powered" trajectories to isolate the benefits of context distillation.
>
> A comparison between context distillation (KL divergence training) and standard cross-entropy supervised fine-tuning is already included in Section 5.4 and Table 7 in Appendix B.2. The results are summarized below:
>
> |Loss|Trajectories|Round|DBLP|Agenda|Yelp|Airbnb|Flight|Coffee|Avg.|
> |-|-|-|-|-|-|-|-|-|-|
> |KL|all|1|92.1|94.4|84.7|89.5|92.2|97.7|91.8|
> |KL|all|2|97.7|92.9|94.9|**96.1**|93.3|**99.5**|95.7|
> |KL|all|3|**98.9**|96.0|**98.9**|**96.1**|**98.3**|**99.5**|**97.9**|
> |cross-entropy (hint-powered SFT)|success only|1|94.9|**99.2**|84.2|88.2|91.1|95.8|92.2|
> |cross-entropy (hint-powered SFT)|success only|2|97.7|96.8|88.7|86.9|93.3|98.6|93.7|
> |cross-entropy (hint-powered SFT)|success only|3|97.7|97.6|97.7|93.5|96.1|97.7|96.7|
>
> We attribute the advantage of KL-based context distillation to **richer supervision from the teacher’s full output distribution** rather than single-token supervision, which in turn makes more efficient use of a limited number of training examples, and increases robustness across training rounds.
>
> In the updated revision, we explicitly highlight this comparison in the main text to make this baseline comparison clearer.
>
> ### **Sensitivity to hint phrasing**
>
> We agree that sensitivity to hint quality or phrasing is an important consideration.
>
> To evaluate our method’s robustness to hint variations, we conducted an experiment in which we asked Llama3.1-70B to rephrase the human-written hints while preserving semantic meaning. The ToolQA Round 1 results are shown below.
>
> The average across ToolQA domains is nearly identical with rephrased hints.
>
> ||DBLP|Agenda|Yelp|Airbnb|Flight|Coffee|Avg.|
> |-|-|-|-|-|-|-|-|
> |ToolQA round1|92.1|94.4|84.7|89.5|92.2|97.7|91.8|
> |w/ rephrased hints|93.2|92.1|85.9|91.5|91.7|97.2|91.9|
>
> As an example of the applied rephrasing, below are a few of the rephrased hints:
>
>
> - Before ending the task, print the final answer to check the value and the format of it. → Prior to completing the task, display the final answer to verify both its value and formatting.
>
> - When a large number of documents are retrieved, it may not be feasible to display all of them at once, as the output could be truncated. → If you retrieve a very large set of documents, displaying them all at once may be impractical since the output is likely to be cut off.
>
> ### **Broader Impact concerns**
> Thank you for mentioning the specific concern of visiblity and auditability of hints. We have added a discussion of this and possible remediating strategies in the Broader impact statement.

---

### Decision · Action_Editor_7odF · 2026-06-22

**Recommendation:** Accept as is

**Additional Comments:**

The authors have addressed the questions and concerns but the reviewers and I thank them for that. The concern for additional experiments is not needed for TMLR in my opinion.

**Audience:**

Yes

**Audience Explanation:**

The paper proposes a method for LLMs to learn complex tasks through hints and feedbacks by a human in the loop and distilling in weights of the LLM. This is of interest to a wide audience.

**Claims And Evidence:**

Yes

**Claims Explanation:**

Current LLMs require comprehensive prompts to perform specialized tasks. As the prompt size increases the compute cost increases and performance may degrade.

The main contribution of the paper is to internalize hints provided by a human expert for a specific step of the task on a recurring basis. The paper reports that after a few iterations such a system will outperform the current method of using prompt based hints.

During the initial training state, prompt tuning is used to generate a set of actions for a variety of tasks. The tasks and outputs, is then used to train a student LLM, which has the initial prompt internalized through LoRA adapter.

In subsequent rounds, an expert analyzes the improved model's results to come up with tasks and action specific hints. The teacher agent is updated with this additional hints and the student is trained on the new output from this updated teacher agent.